# Joint Implicit Neural Representations for Global-Scale Species Mapping

## Abstract

Estimating the geographical range of a species from sparse observations is a challenging and important geospatial prediction problem. Given a set of locations where a species has been observed, the goal is to learn a model that can predict whether the species is present or absent at any location. This problem has a long history in ecology, but traditional methods struggle to take advantage of emerging large-scale crowdsourced datasets which can include tens of millions of records for hundreds of thousands of species. We propose a new approach based on implicit neural representations that jointly estimates the geographical ranges of 47k species simultaneously. We also introduce a series of benchmarks that measure different aspects of species range estimation and spatial representation learning. We find that our approach scales gracefully, making better predictions as we scale up the number of species used for training and the amount of training data per species. Despite being trained on noisy and biased crowdsourced data, our models can approximate expert-developed gold standard range maps for many species.

## 1 Introduction

We are witnessing a dramatic decline in global biodiversity, which has severe ramifications for natural resource management, food security, and ecosystem services that are crucial to human health (Watson et al., 2019). If we want to take effective conservation action we must understand where different species live. In ecology, this problem is known as *species distribution modeling* (SDM) (Elith & Leathwick, 2009). Ideally we would have up-to-date global SDMs for all species. Unfortunately, we only have SDMs for a relatively small number of species and locations.

A key obstacle is that most SDM methods are incompatible with the most common form of species data. An SDM must predict whether a species is present or absent at any location given spatially sparse observation records. With a sufficient amount of *presence-absence* data – records of where a species has been observed to occur and where it has been confirmed to be absent – this problem can be approached using standard methods from machine learning and statistics (Beery et al., 2021).[1] Most SDM methods require presence-absence data. Unfortunately, presence-absence data is scarce due to the difficulty of verifying that a species is absent from an area. *Presence-only* data – which consists only of locations where a species has been observed, with no confirmed absences – is much more abundant. For instance, at the end of 2022 the iNaturalist community science platform (iNa) had collected over 70M observations across 300k species, all of which is presence-only data. Though presence-only data is not without drawbacks (Hastie & Fithian, 2013), it is important to develop SDM methods that can take advantage of this vast supply of data.

Deep learning is currently one of our best tools for making use of large-scale datasets. Deep neural networks also have a key advantage over many prior SDM methods because they can *jointly* learn the distribution of many species in the same model (Chen et al., 2017; Tang et al., 2018; Mac Aodha et al., 2019). By learning representations that *share* information across species, the models can make improved predictions (Chen et al., 2017). However, the majority of these deep learning approaches need presence-absence data for training, which prevents them from scaling beyond the small number of species and regions for which sufficient presence-absence data is available.

---

[1] The term "presence-absence" should not be taken to convey absolute certainty about whether a species is present or absent. False absences and, to a lesser extent, false presences are a serious concern in SDM (MacKenzie et al., 2002; Guillera-Arroita, 2017).

Figure 1: We show that sparse species observation data *(left)* can be used to learn meaningful geospatial representations *(middle)*. We evaluate the ability of these models to *(right)*: estimate species ranges, assist computer vision classifiers, and transfer to other geospatial prediction tasks.

Our work makes the following contributions: (i) A novel implicit neural representation approach to joint SDM across tens of thousands of species, trained with crowdsourced presence-only data. (ii) A detailed investigation of loss functions for learning from presence-only data, their scaling properties, and the resulting geospatial representations. (iii) A new suite of geospatial benchmark tasks – ranging from species mapping to fine-grained visual categorization – which will facilitate future research on large-scale SDM and geospatial representation learning.

## 2 RELATED WORK

Species distribution modeling (SDM) refers to a set of methods that aim to predict where (and sometimes when and in what quantities) species of interest are likely to be found (Elith & Leathwick, 2009). The literature on SDM is vast. Readers interested in an overview of SDM should see the classic review by Elith & Leathwick (2009) or the recent review of SDM for computer scientists by Beery et al. (2021). Note that we focus narrowly on the problem of estimating species range, i.e. we do not consider more complex problems like abundance estimation (Potts & Elith, 2006).

Traditional approaches to SDM train conventional supervised learning models (e.g. logistic regression (Pearce & Ferrier, 2000), random forests (Cutler et al., 2007), etc.) to learn a mapping between hand-designed sets of environmental features (e.g. altitude, average rainfall, etc.) and species presence or absence (Phillips et al., 2004; Elith et al., 2006). Readers interested in traditional SDM approaches should consult Norberg et al. (2019); Valavi et al. (2021; 2022) and the references therein. More recently, deep learning methods have been introduced that instead *jointly* represent multiple different species within the same model (Chen et al., 2017; Botella et al., 2018b; Tang et al., 2018; Mac Aodha et al., 2019). These models are typically trained on crowdsourced data, which can introduce additional challenges and biases that need to be accounted for during training (Fink et al., 2010; Chen & Gomes, 2019; Johnston et al., 2020; Botella et al., 2021). We build on the work of Mac Aodha et al. (2019), who proposed a neural network approach that forgoes the need for environmental features (as required by e.g. Botella et al. (2018b); Tang et al. (2018)) by learning to predict species presence from geographical location alone.

The problem of joint SDM with presence-only data can be viewed as an instance of multi-label classification with limited supervision. In particular, it is an example of single positive multi-label learning (SPML) (Cole et al., 2021; Verelst et al., 2022; Zhou et al., 2022). The goal is to train a model that is capable of making multi-label predictions at test time, despite having only ever observed one positive label per training instance (i.e. no confirmed negative training labels). Our work connects the SPML literature and species range mapping literature, and sets up large-scale joint species distribution modeling as a challenging real-world SPML task. This setting presents significant new challenges for SPML, which has previously been limited to relatively small label spaces ($< 100$ categories). We investigate the role of the number of categories in our experiments. Some SPML methods such as ROLE (Cole et al., 2021) are not computationally viable when the label space is large. One of our baselines will be the SPML method of Zhou et al. (2022), which is scalable and attains state-of-the-art performance on standard SPML benchmarks.

Our task is related to the growing number of works that use coordinate neural networks for implicitly representing images (Tancik et al., 2020) and 3D scenes (Sitzmann et al., 2019), including those that perform novel-view image synthesis (Mildenhall et al., 2020). There are many design choices in these methods that are being actively studied, including the impact of the activation functions in the network (Sitzmann et al., 2019; Ramasinghe & Lucey, 2022) and the effect of different input encodings (Tancik et al., 2020; Zheng et al., 2022). In most research on implicit neural representa-

tions, there is a natural choice of training objective e.g. mean squared error between the predictions and the data. In the context of presence-only species range estimation, this choice is less clear. We systematically investigate this question in our experiments.

Quantifying the performance of SDMs at scale is notoriously difficult due to the fact that we lack presence-absence data for most species and locations (Beery et al., 2021). One approach is to evaluate performance on a small set of species (typically fewer than 300) from limited geographical regions where it is feasible to collect presence-absence data, as done in e.g. Potts & Elith (2006); Norberg et al. (2019); Valavi et al. (2022). Two of our evaluation tasks are larger-scale versions of this idea, in which we compare the performance of our models against gold standard expert range maps for around 4,000 species. An alternative evaluation approach is to measure the performance on a related "proxy" task. For example, there have been a number of works that use models trained for species range estimation to assist deep image classifiers (Berg et al., 2014; Tang et al., 2015; Mac Aodha et al., 2019; Chu et al., 2019; Mai et al., 2020; Terry et al., 2020; Skreta et al., 2020; Yang et al., 2022). By using images from platforms like iNaturalist, we can evaluate different range estimation methods on the task of aiding fine-grained image classification across tens of thousands of species. Finally, we also evaluate the spatial representations learned by our models via transfer learning, using them as inputs for a suite of geospatial regression tasks. This suite of complementary benchmark tasks captures different aspects of SDM performance, and provides a starting point for large-scale SDM evaluation. See Appendix B.1 for a more detailed comparison of approaches to SDM benchmarking.

## 3 METHODS

### 3.1 PRELIMINARIES

**Problem statement.** Let $\mathbf{x} = [lon, lat]$ denote a geographical location (i.e. longitude and latitude). Let $\mathbf{y} \in \{0, 1\}^S$ denote the true presence (1) or absence (0) of $S$ different species at $\mathbf{x}$. We introduce $\mathbf{z} \in \{0, 1, \varnothing\}^S$ to represent our observed data at $\mathbf{x}$, where $z_j = 1$ if species $j$ is present, $z_j = 0$ if species $j$ is absent, and $z_j = \varnothing$ if we do not know whether species $j$ is present or absent. Our goal is to develop a model that produces an estimate of $\mathbf{y}$ at any location $\mathbf{x}$ over some spatial domain $\mathcal{X}$, given observed data $\{(\mathbf{x}_i, \mathbf{z}_i)\}_{i=1}^N$. We parameterize this model as $\hat{\mathbf{y}} = h_\phi(f_\theta(\boldsymbol{x}))$, where $f_\theta : \mathcal{X} \to \mathbb{R}^k$ is a location encoder and $h_\phi : \mathbb{R}^k \to [0, 1]^S$ is a multi-label classifier. The prediction $\hat{\boldsymbol{y}} \in [0, 1]^S$ is our estimate of the presence or absence of each species at $\mathbf{x}$. Intuitively, the location encoder $f_\theta$ learns a representation of space that is used by the multi-label classifier $h_\phi$ to predict species presence at a given location.

**Input encoding.** Each species observation is associated with a point $\mathbf{x} = [lon, lat]$. We rescale these values so that $lon, lat \in [-1, 1]$. Following Mac Aodha et al. (2019) we guard against boundary effects by using a sinusoidal encoding for the input. The final result is an input vector

$$\mathbf{x} = [\sin(lon), \cos(lon), \sin(lat), \cos(lat)]. \tag{1}$$

Alternative input encodings for related coordinate networks have been explored in existing literature (Mai et al., 2020; Tancik et al., 2020; Mai et al., 2022; Zheng et al., 2022). Their choice is orthogonal to the losses we explore, thus we leave their evaluation for future work.

**Implicit neural representations.** Traditionally, representation learning aims to transform complex objects (e.g. images, text) into simpler objects (e.g. low-dimensional vectors) that facilitate some downstream task (e.g. classification) (Goodfellow et al., 2016). Implicit neural representations offer a different perspective, in which a signal is represented by a neural network that maps the signal domain (e.g. $\mathbb{R}$ for audio, $\mathbb{R}^2$ for images) to the signal values (Sitzmann et al., 2019; Tancik et al., 2020). In this work we learn an implicit neural representation for a large collection of crowdsourced species observation data, which can be viewed as a set of 2D point clouds (one for each species). This process yields an implicit neural representation for the geospatial distribution of each species, as well as a representation for any location on earth.

**Presence-absence vs. presence-only data.** Species observation datasets come in two varieties: (i) *Presence-absence* data consists of locations where a species has been observed to be present and locations where it has been confirmed to be absent. That is, we say we have presence-absence data for species $j$ if $|\{\mathbf{z}_i : z_{ij} = 0\}| > 0$ and $|\{\mathbf{z}_i : z_{ij} = 1\}| > 0$. Unfortunately, presence-absence data

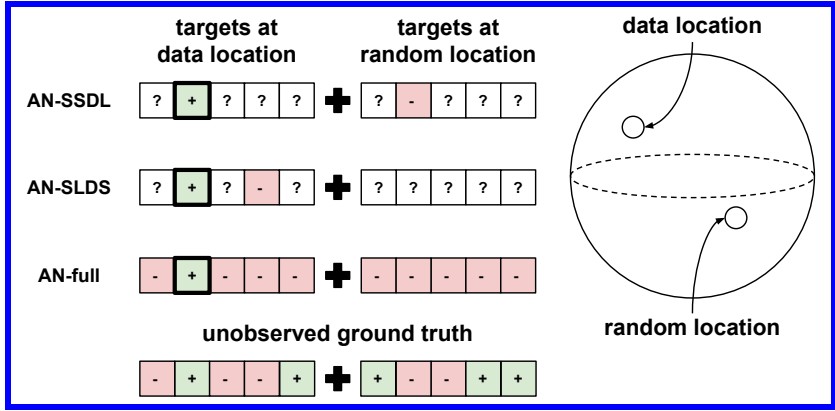

Figure 2: Illustration of three basic losses from Section 3.2. For each loss ($\mathcal{L}_{\mathrm{AN-SSDL}}$, $\mathcal{L}_{\mathrm{AN-SLDS}}$, and $\mathcal{L}_{\mathrm{AN-full}}$) we visualize the targets that the network is encouraged to predict. Each loss can be broken into two parts: one part that updates the network's predictions at the location of a training example (the *data location*) and one part that updates the network's predictions at another location chosen randomly (*random location*). Each loss has access to one confirmed positive label (bold boxes). The rest of the labels are unobserved (non-bold boxes), and the losses make different assumptions about those unobserved labels. $\mathcal{L}_{\mathrm{AN-SSDL}}$ (top) assumes that the species that has been confirmed present at the data location is absent from the random location. $\mathcal{L}_{\mathrm{AN-SLDS}}$ (middle) assumes that a random species (other than the one that is confirmed to be present) is absent from the data location. $\mathcal{L}_{\mathrm{AN-full}}$ (bottom) assumes that all unobserved species are absent from the data location, and that all species are absent from the random location. All of these losses are making imperfect approximations of the unobserved ground truth labels shown at the bottom of the figure.

is costly to obtain at scale because confirming absence requires skilled observers to exhaustively search an area. (ii) *Presence-only* data is easier to acquire and thus more abundant because absences are not collected, i.e. $z_{ij} \in \{1, \varnothing\}$ for $i \in [N]$ and $j \in [S]$.

## 3.2 LEARNING FROM LARGE-SCALE PRESENCE-ONLY DATA

In the context of SPML *image* classification, a simple but effective approach is to assume that unobserved labels are negative (Cole et al., 2021). This approach is based on a probabilistic argument: since natural images tend to have a small number of categories compared to the size of the label set, the vast majority of the labels will be negative. This is also true for species distribution modeling. Given an arbitrary location and a large set of candidate species, nearly all of them will be absent. In this section we describe several simple and scalable loss functions based on this idea. We illustrate three of our basic losses in Figure 2.

**"Assume negative" loss (same species, different location).** This loss pairs each observation of a species with a pseudo-negative for that species at another location chosen uniformly at random:

$$\mathcal{L}_{\mathrm{AN-SSDL}}(\hat{\mathbf{y}}, \mathbf{z}) = -\frac{1}{n_{\mathrm{pos}}} \sum_{j=1}^{S} \mathbb{1}_{[z_j=1]}[\log(\hat{y}_j) + \log(1 - \hat{y}_j')] \tag{2}$$

where $\hat{\mathbf{y}}' = h_\phi(f_\theta(\mathbf{r}))$ with $\mathbf{r} \sim \mathrm{Uniform}(\mathcal{X})$ and $n_{\mathrm{pos}} = \sum_{j=1}^{S} \mathbb{1}_{[z_j=1]}$. This approach generates pseudo-negatives across the globe, but many of them are likely to be "easy" because they are in the ocean or far from the true species range.

**"Assume negative" loss (same location, different species).** This loss pairs each observation of a species with a pseudo-negative at the same location for a different species:

$$\mathcal{L}_{\mathrm{AN-SLDS}}(\hat{\mathbf{y}}, \mathbf{z}) = -\frac{1}{n_{\mathrm{pos}}} \sum_{j=1}^{S} \mathbb{1}_{[z_j=1]}[\log(\hat{y}_j) + \log(1 - \hat{y}_{j'})] \tag{3}$$

where $j' \sim \mathrm{Uniform}(\{j : z_j \neq 1\})$. Intuitively, this approach generates pseudo-negatives that are aligned with the spatial distribution of the observed data.

**Full "assume negative" loss.** The previous two losses are inefficient in the sense that they do not use all of the entries in $\hat{\mathbf{y}}$. We can combine the pseudo-negative sampling strategies of $\mathcal{L}_{\text{AN}-\text{SSDL}}$ and $\mathcal{L}_{\text{AN}-\text{SLDS}}$ and use all available predictions as follows:

$$\mathcal{L}_{\text{AN}-\text{full}}(\hat{\mathbf{y}}, \mathbf{z}) = \frac{1}{S} \sum_{j=1}^{S} \left[ \mathbb{1}_{[z_j=1]} \lambda \log(\hat{y}_j) + \mathbb{1}_{[z_j \neq 1]} \log(1 - \hat{y}_j) + \log(1 - \hat{y}_j') \right] \qquad (4)$$

where $\hat{\mathbf{y}}' = h_\phi(f_\theta(\mathbf{r}))$ with $\mathbf{r} \sim \text{Unif}(\mathcal{X})$. The hyperparameter $\lambda > 0$ can be used to prevent the negative labels from dominating the loss. This is equivalent to the loss from Mac Aodha et al. (2019) without their user modeling terms.

**Maximum entropy loss.** The current state-of-the-art in SPML image classification is a technique proposed by Zhou et al. (2022). The idea is to encourage predictions for unobserved labels to maximize entropy instead of forcing them to zero. Zhou et al. (2022) also includes an additional pseudo-labeling component, but we omit this because the resulting empirical improvement from it is small. We can apply this idea to $\mathcal{L}_{\text{AN}-\text{SSDL}}$, $\mathcal{L}_{\text{AN}-\text{SLDS}}$, and $\mathcal{L}_{\text{AN}-\text{full}}$ by replacing all terms of the form "$-\log(1-p)$" with terms of the form "$H(p)$" where $H(p) = -(p \log(p) + (1-p) \log(1-p))$. We name these "maximum entropy" (ME) variants $\mathcal{L}_{\text{ME}-\text{SSDL}}$, $\mathcal{L}_{\text{ME}-\text{SLDS}}$, and $\mathcal{L}_{\text{ME}-\text{full}}$.

## 4 EXPERIMENTS

In this section we evaluate the performance of the different losses outlined previously across a range of species and environmental prediction tasks.

### 4.1 MODELS

As described in Section 3.1, our model consists of a location encoder $f_\theta$ and a multi-label classifier $h_\phi$ which produce a vector of predictions $\hat{y} = h_\phi(f_\theta(\mathbf{x}))$ for a location $\mathbf{x}$. The location encoder $f_\theta$ is implemented as the fully connected neural network shown in Figure A4. We implement the multi-label classifier $h_\phi$ as a single linear layer with sigmoid activations. Full implementation details can be found in Appendix C.

There are two other model types represented in Table 1. The rows tagged with "LR" are logistic regression baselines (Pearce & Ferrier, 2000), in which the location encoder $f_\theta$ is replaced with the identity function and $h_\phi$ is unchanged. The "Best Discretized Grid" row does not use a location encoder, but instead makes predictions based on binning the training data (Berg et al., 2014). Full details for these models can be found in Appendix C. These baselines allow us to quantify the importance of our deep location encoder.

### 4.2 TRAINING DATA

We train our models on presence-only species observation data obtained from the community science platform iNaturalist (iNa). The training set consists of 35.5 million observations covering 47,375 species observed prior to 2022. For each species, we have the geographical coordinate where the species was observed. To be included in the training set, each species needed at least 50 observations. Some species are more common than others, and thus the dataset is heavily imbalanced – see Figure A6. Later we use this data in its entirety ("All"), with different maximum observations per class ("X / Class"), or with different subsets of classes. See Appendix D for more details on the dataset.

### 4.3 EVALUATION TASKS AND METRICS

We propose four tasks for evaluating large-scale species distribution models. We give brief descriptions here, and provide further details in Appendix E.

**S&T: eBird Status and Trends.** This task quantifies the agreement between our presence-only maps and high-quality expert range maps from the *eBird Status & Trends* dataset (Fink et al., 2020), covering 536 bird species. Performance is measured using mean average precision (MAP), i.e. computing the per-species average precision (AP) and averaging the results across species.

**IUCN: Range Maps.** This task compares our maps against expert range maps from the International Union for Conservation of Nature (IUCN) Red List (IUC). Unlike the bird-focused *S&T* task, this task covers 3,938 species from diverse taxonomic groups. Performance is measured using MAP.

**Geo Prior: Geographical Priors for Computer Vision.** This task measures the utility of our range maps as priors for fine-grained image classification (Berg et al., 2014; Mac Aodha et al., 2019). For this task we collect 282,974 geo-located images from iNaturalist, covering 39,444 species from our training set. The performance metric for this task ("$\Delta$ Top-1") is the change in image classifier top-1 accuracy when using our range maps as a geographical prior. This occurs at test time – the image classifier is never trained with geographical information. A positive value indicates that the prior improves the performance of the classifier. Unlike *S&T* and *IUCN*, this is an *indirect* evaluation of range map quality since we are assessing only how useful the range maps are for a downstream task.

**Geo Feature: Environmental Representation Learning.** Instead of evaluating the range estimates of our model, this transfer learning task evaluates the quality of the underlying geospatial representation that has been learned. The goal is to predict nine different geospatial characteristics of the environment e.g. above-ground carbon, elevation, etc. First, we use the location encoder $f_\theta$ to extract features for a grid of evenly spaced locations. After splitting the locations into train and test sets, we fit a linear regressor to predict the geospatial characteristics from the extracted features. Performance is evaluated using the coefficient of determination $R^2$ on the test set.

## 4.4 RESULTS

**Deep location encoders are crucial.** Our method outperforms logistic regression trained with the same inputs and loss by more than 50 MAP on the *S&T* task – compare "$\mathcal{L}_{AN-full}$" and "$\mathcal{L}_{AN-full}$ (LR, Coord. Inputs)" with 1000 examples per class in Table 1. The only difference is the fact that our method uses a deep location encoder. See Appendix A.2 for more details.

**Environmental features are unnecessary for good performance.** Our method gains only 1.1 MAP on the *S&T* task when we switch from coordinate inputs ("$\mathcal{L}_{AN-full}$") to environmental feature inputs ("$\mathcal{L}_{AN-full}$ (Env. Inputs)"). The gain is much more significant for logistic regression, i.e. +15.7 MAP for "$\mathcal{L}_{AN-full}$ (LR, Env. Inputs)" vs. "$\mathcal{L}_{AN-full}$ (LR, Coord. Inputs)". It seems that our approach can successfully use sparse presence-only data to learn about the environment, so that using environmental features as input provides a marginal benefit. See Appendix A.1 for more.

**Low-shot performance is surprisingly good.** In Table 1 we can see that $\mathcal{L}_{AN-full}$ (using only 10 examples per category, i.e. 1.3% of the training data) beats the "Best Discretized Grid" baseline (which uses all of the training data) on every task. Our models seem to be capable of capturing general spatial patterns using relatively little data. While this is encouraging, we expect that more data is necessary to capture fine detail as suggested by Figure 5.

**Pseudo-negatives that follow the data distribution are usually better.** $\mathcal{L}_{AN-SSDL}$ and $\mathcal{L}_{AN-SLDS}$ differ only in the fact that $\mathcal{L}_{AN-SSDL}$ samples pseudo-negatives from random locations while $\mathcal{L}_{AN-SLDS}$ samples pseudo-negatives from data locations (see Figure 2). We can see in Table 1 that $\mathcal{L}_{AN-SLDS}$ outperforms $\mathcal{L}_{AN-SSDL}$ for all tasks except *IUCN*. This could be due to the fact that some *IUCN* species have ranges far from areas that are well-sampled by iNaturalist. As we can see in Figure A3 (Black Oystercatcher), $\mathcal{L}_{AN-SSDL}$ can behave poorly in areas with little training data. This highlights the importance of using diverse tasks to study range estimation methods.

**Joint learning across categories is beneficial, but more data is better.** In Figure 3 we study the effect of the amount of training data on performance for the *S&T* task. We first note that increasing the number of training examples per species reliably and significantly improves performance. One possible mechanism for this is suggested by Figure 5, which shows a more spatially detailed representation emerging with more training data. Figure 3 also shows that adding additional species to the training set (which are not in the test set) also tends to increase performance, though by a smaller margin. In the right panel of Figure 3 we can see that it is better to have more training data for fewer species than less training data spread across more species. Table 1 shows a clear trend towards higher performance with more training data across tasks. The exception is that both $\mathcal{L}_{AN-SLDS}$ and $\mathcal{L}_{AN-full}$ have worse performance on the *Geo Prior* task when using all data compared to using 1000 / class. This may be due to the extreme imbalance in the full training set, as show in Figure A6.

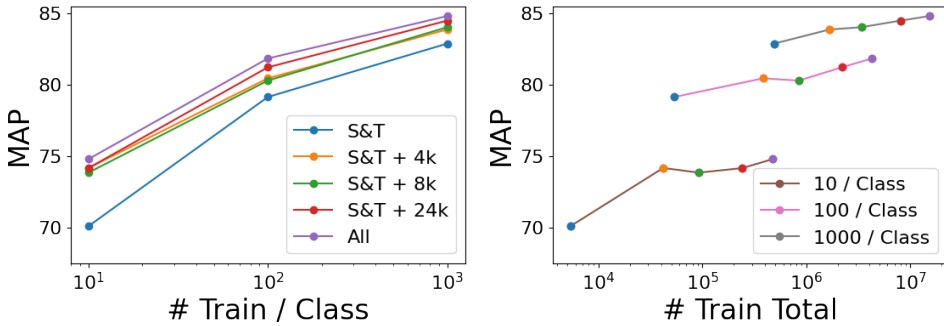

Figure 3: S&T range prediction performance with $\mathcal{L}_{\text{AN-full}}$ as a function of training set size. We consider two notions of size: number of training examples per class and number of classes (i.e. species). These plots visualize the same data in different ways. *(Left)* Each curve corresponds to a model trained on a different set of species. "S&T" indicates that we only train on the species in the S&T task. For "S&T + X" we add in X species chosen uniformly at random. For the "All" line we train on all 47k species. The $x$-axis represents the number of training examples per class. *(Right)* Each curve corresponds to a different amount of training data per class. The points along each curve correspond to the different training sets from the left panel (i.e. "S&T", "S&T + 4k", ..., "All"). The $x$-axis represents the total number of training examples. Unsurprisingly, increasing the number of training examples per category improves performance. More interestingly, adding training data for additional species (which are not evaluated at test time) improves performance as well.

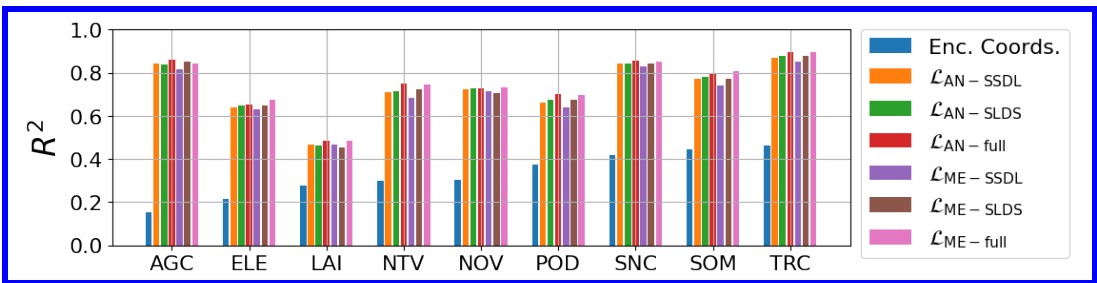

Figure 4: *Geo Feature* task performance broken down by subtask. All location encoders are trained with 1000 examples per category. The regression tasks are sorted by the performance of "Enc. Coords." which is a naive baseline method that only uses the raw encoded coordinates from Equation 1 as features. All losses outperform this naive baseline, but no loss consistently outperforms the others.

**Method rankings may not generalize across domains.** The presence-only SDM problem in this work and the SPML image classification problem in Cole et al. (2021) are both SPML problems. Despite this formal equivalence, it does not seem to be the case that the best current method for SPML image classification is also the best methods for presence-only SDM. Zhou et al. (2022) show that their "maximum entropy" loss performs much better than the "assume negative" loss across a number of image classification datasets. However, all of the "maximum entropy" losses in Table 1 ($\mathcal{L}_{\text{ME-SSDL}}$, $\mathcal{L}_{\text{ME-SLDS}}$, $\mathcal{L}_{\text{ME-full}}$) perform worse than their "assume negative" counterparts ($\mathcal{L}_{\text{AN-SSDL}}$, $\mathcal{L}_{\text{AN-SLDS}}$, $\mathcal{L}_{\text{AN-full}}$) in almost all cases. *$\mathcal{L}_{\text{ME-full}}$ is a top performer on the Geo Feature task, but Figure 4 shows that no method seems to have a commanding advantage.*

### 4.5 LIMITATIONS

There are some limitations associated with our analysis. As noted, the training set is heavily imbalanced, both in terms of the species themselves and where the data was collected. This is partially because some species are more common and thus more likely to be observed than others. We do not explicitly deal with species imbalance in the training data, other than by showing that the ordering of methods does not significantly vary even when the training data is capped to the same upper limit per species, i.e. 10, 100, or 1,000 examples per class in Table 1.

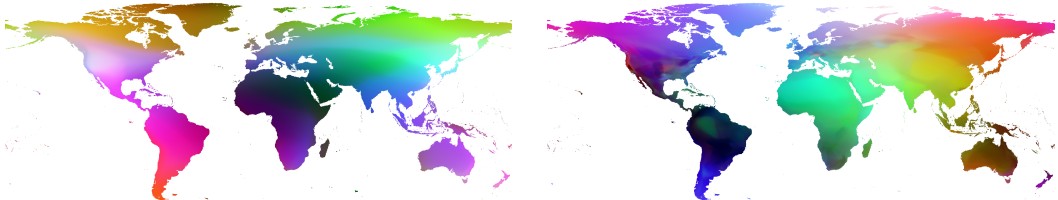

Figure 5: Visualization of the 256-dimensional learned location encoder features projected to three dimensions using Independent Component Analysis. *(Left)* This corresponds to the $\mathcal{L}_{\text{AN}-\text{full}}$ model with a maximum of *10* examples per class. The features are smooth and do not appear to encode much high frequency spatial information. *(Right)* In contrast, the $\mathcal{L}_{\text{AN}-\text{full}}$ model with a maximum of *1000* examples per class, contains more high frequency information. The increase in training data appears to enable this model to better encode spatially varying environmental properties.

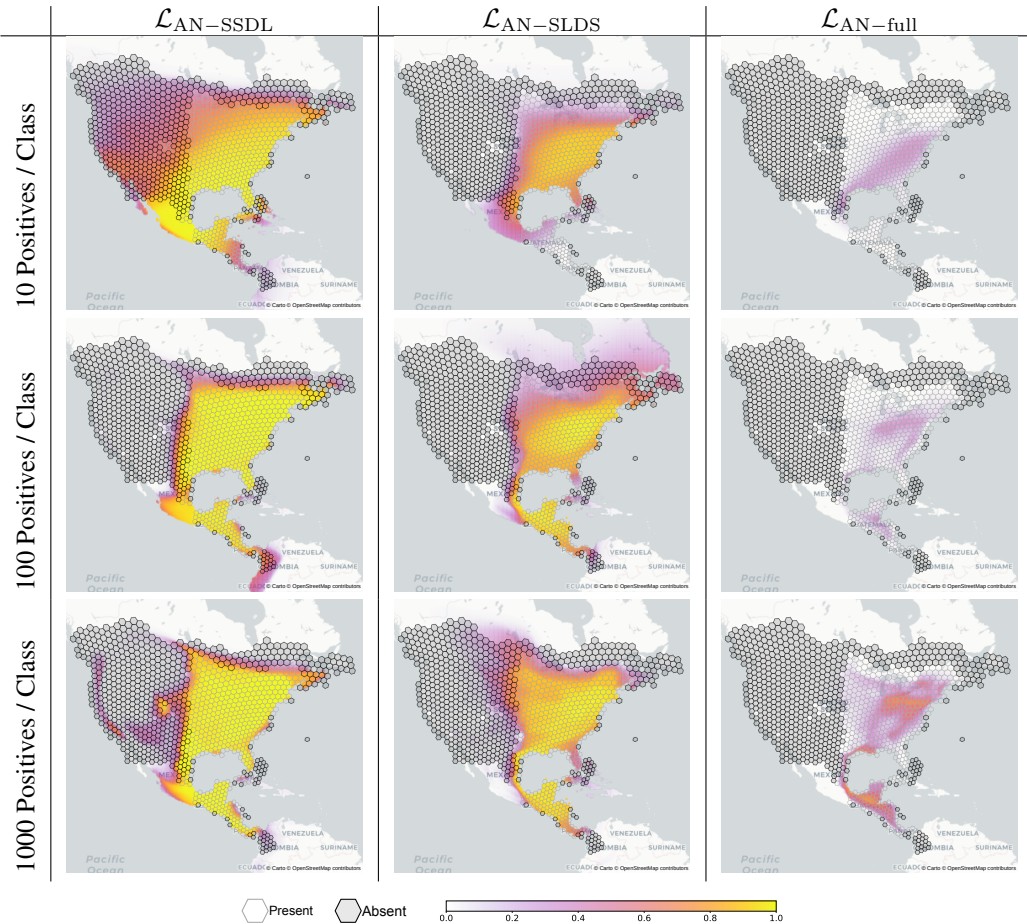

Figure 6: Visualization of predictions for Wood Thrush when varying the amount of training data (rows) for different loss functions (columns). The model predictions are generated on a grid of coordinates. Hexagons signify locations where we can evaluate the model outputs using S&T data. Light hexagons are locations where the model should predict present, dark hexagons are locations where the model should predict absent, and locations without hexagons are not used for S&T evaluation. All models improve their range maps when given access to more data, as expected. $\mathcal{L}_{\text{AN}-\text{SSDL}}$ over estimates the range with few examples, and refines the range with additional data, while $\mathcal{L}_{\text{AN}-\text{full}}$ starts off conservative with few training examples and expands as more data is available. The range predicted by $\mathcal{L}_{\text{AN}-\text{SLDS}}$ is somewhere in-between. The magnitude of the score values is also interesting, with both $\mathcal{L}_{\text{AN}-\text{SSDL}}$ and $\mathcal{L}_{\text{AN}-\text{SLDS}}$ predicting more locations confidently than $\mathcal{L}_{\text{AN}-\text{full}}$.

Table 1: Results for four geospatial tasks: **S&T** (eBird Status & Trends species mapping), **IUCN** (IUCN species mapping) , **Geo Prior** (fine-grained image classification with a geographical prior), and **Geo Feature** (geographical feature regression). The tasks and metrics are defined in Section 4.3. We assess performance as a function of the loss function and the amount of training data ("# / Class"). The logistic regression ("LR") and "Best Discretized Grid" baselines do not have an entry for the Geo Feature task as they do not learn a location encoder. "$\mathcal{L}_{\text{AN−full}}$ (Env. Inputs)" does not have an entry for the Geo Feature task because it is trained on environmental features which may be closely related to the evaluation task.

| Loss | # / Class | S&T (MAP) | IUCN (MAP) | Geo Prior ($\Delta$ Top-1) | Geo Feature (Mean $R^2$) |
|---|---|---|---|---|---|
| $\mathcal{L}_{\text{AN−SSDL}}$ | 10 | 65.65 | 95.94 | +3.6 | 0.671 |
| $\mathcal{L}_{\text{AN−SSDL}}$ | 100 | 73.54 | 97.54 | +4.7 | 0.731 |
| $\mathcal{L}_{\text{AN−SSDL}}$ | 1000 | 76.52 | 98.02 | +4.8 | 0.725 |
| $\mathcal{L}_{\text{AN−SSDL}}$ | All | 77.35 | 98.11 | +4.9 | 0.731 |
| $\mathcal{L}_{\text{AN−SLDS}}$ | 10 | 72.09 | 92.74 | +4.8 | 0.705 |
| $\mathcal{L}_{\text{AN−SLDS}}$ | 100 | 79.66 | 95.06 | +6.2 | 0.724 |
| $\mathcal{L}_{\text{AN−SLDS}}$ | 1000 | 82.68 | 94.45 | +6.2 | 0.729 |
| $\mathcal{L}_{\text{AN−SLDS}}$ | All | 82.98 | 95.36 | +5.9 | 0.744 |
| $\mathcal{L}_{\text{AN−full}}$ | 10 | 74.78 | 96.70 | +4.5 | 0.699 |
| $\mathcal{L}_{\text{AN−full}}$ | 100 | 81.83 | 97.78 | +6.6 | 0.735 |
| $\mathcal{L}_{\text{AN−full}}$ | 1000 | 84.80 | 98.27 | +6.2 | 0.746 |
| $\mathcal{L}_{\text{AN−full}}$ | All | 85.15 | 98.19 | +5.0 | 0.743 |
| $\mathcal{L}_{\text{AN−full}}$ (Env. Inputs) | 1000 | 85.94 | - | - | - |
| *Baselines:* | | | | | |
| $\mathcal{L}_{\text{AN−full}}$ (LR, Coord. Inputs) (Pearce & Ferrier, 2000) | 10 | 31.46 | 49.25 | -0.4 | - |
| $\mathcal{L}_{\text{AN−full}}$ (LR, Coord. Inputs) (Pearce & Ferrier, 2000) | 100 | 30.05 | 44.95 | -0.3 | - |
| $\mathcal{L}_{\text{AN−full}}$ (LR, Coord. Inputs) (Pearce & Ferrier, 2000) | 1000 | 32.14 | 39.49 | -0.6 | - |
| $\mathcal{L}_{\text{AN−full}}$ (LR, Env. Inputs) (Pearce & Ferrier, 2000) | 1000 | 47.83 | - | - | - |
| $\mathcal{L}_{\text{ME−SSDL}}$ (Zhou et al., 2022) | 1000 | 72.48 | 96.56 | +1.6 | 0.708 |
| $\mathcal{L}_{\text{ME−SLDS}}$ (Zhou et al., 2022) | 1000 | 81.86 | 93.62 | +2.1 | 0.728 |
| $\mathcal{L}_{\text{ME−full}}$ (Zhou et al., 2022) | 1000 | 82.52 | 94.71 | +1.5 | 0.747 |
| Best Discretized Grid (Berg et al., 2014) | All | 72.87 | 94.42 | +4.1 | - |
| $\mathcal{L}_{\text{GP}}$ (Mac Aodha et al., 2019) | 1000 | 81.97 | 97.73 | +5.7 | 0.718 |

Reliably evaluating the performance of models for species range estimation is a long standing and challenging question. To address this issue, we present a suite of complementary benchmarks that attempt to evaluate different facets of this spatial prediction problem. However, obtaining ground truth distribution data for tens of thousands of species remains a very difficult problem. While we believe our benchmarks to be a significant step forward, they are likely to have blind spots.

Finally, extreme care should be taken before making conservation decisions based on the outputs of models such as the ones presented here. Our goal in this work is to demonstrate the promise of large-scale representation learning for species distribution modeling. Our models have not been calibrated or validated beyond the experiments illustrated above.

## 5 CONCLUSION

We explored the problem of species range estimation through the lens of representation learning. Across a range of spatial prediction tasks, we showed that adding more and diverse data improves model performance. In doing so, we also made connections to recent literature on training implicit coordinate networks and learning multi-label classifiers from limited supervision. We hope that our work will encourage more machine learning researchers to work on this important problem.

While the initial results are encouraging, there are still many avenues for future work. For example, currently our models do not make use of any temporal input information (Mac Aodha et al., 2019), we do not account for the spatial biases present in the data (Chen & Gomes, 2019), and our neural network models could be adapted to have better inductive biases for encoding spatially varying signals (Ramasinghe & Lucey, 2022).

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

# Appendix

## A    ADDITIONAL RESULTS

### A.1    HOW DOES OUR APPROACH COMPARE TO A TRADITIONAL SDM?

Our approach can handle large-scale SDM in a way that most traditional SDMs cannot. However, for context we compare our results to a classic SDM. One of the most scalable SDM approaches in the ecology literature is to use logistic regression with environmental features as input (Pearce & Ferrier, 2000). Crucially, we can train this method using GPU-accelerated batch-based optimization. Full implementation details are provided in Appendix C.3.1.

In Figure A1 we compare our approach against logistic regression. We try both approaches with environmental feature inputs and with coordinate inputs. First we note that our method has a decisive advantage over logistic regression in all cases. Second, we observe that using environmental features as input (instead of coordinates) improves the performance of logistic regression significantly (+15.7 MAP) and improves the performance of our method slightly (+1.1 MAP). These facts are consistent with the claim that our method is able to learn a rich representation of the environment from species co-occurrence patterns alone, without the need for environmental covariates.

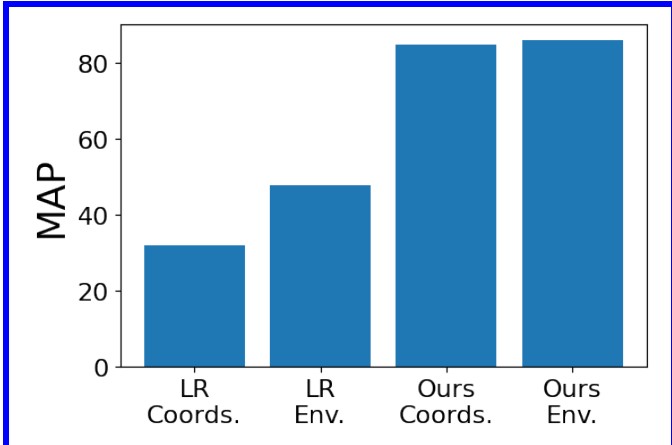

Figure A1:    Results for the *S&T* task. All models are trained with 1000 examples per class using the $\mathcal{L}_{\mathrm{AN-full}}$ loss. We compare logistic regression ("LR") to our approach ("Ours"), using either coordinates ("Coords.") or environmental covariates ("Env.") as inputs.

### A.2    WHAT IS THE EFFECT OF THE ARCHITECTURE OF THE LOCATION ENCODER?

**Location encoder depth.** As discussed in Appendix C, the location encoder $f_\theta$ used throughout the paper has one input layer followed by four residual layers. We now consider the importance of the number of layers in the encoder. Figure A2 shows a large performance increase when we go from the logistic regression model (0 hidden layers) to the next largest model (4 hidden layers). Performance is fairly stable beyond that point, marginally increasing (8 hidden layers) and then dropping (12 hidden layers). We suggest that the performance drop for the deepest models may be due to the fact that, with enough capacity, there is no need to share information across species.

**Residual modules vs. traditional layers.** The residual network used by us was chosen to allow us to directly compare to existing related work that uses the same architecture (Mac Aodha et al., 2019). Figure A2 compares this architecture against a simpler multi-layer perceptron (MLP) architecture. Here the traditional MLP is comparable, and in fact performs slightly better. Though there is not much difference in performance, future work should consider using a traditional MLP for simplicity.

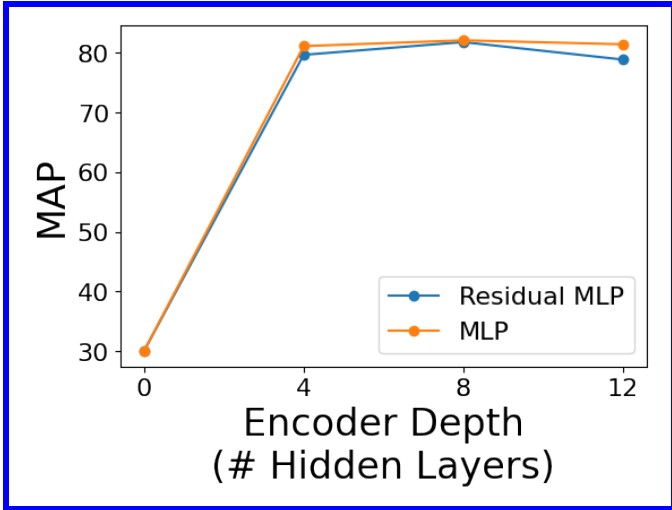

Figure A2: Results for the *S&T* task under different design choices for the location encoder architecture. All models are trained with 100 examples per class. We compare the location encoder architecture used throughout the paper ("Residual MLP") against a simple MLP as a function of the number of layers in the encoder. Note that one residual layer corresponds to two traditional layers, so for the purposes of this figure the encoder used throughout the paper has 8 layers (8 ordinary layers = 4 residual layers). The 0-layer model is identical for both architectures, and is equivalent to logistic regression.

### A.3 ADDITIONAL QUALITATIVE RESULTS

To build some intuition for the behavior of $\mathcal{L}_{AN-SSDL}$, $\mathcal{L}_{AN-SLDS}$, and $\mathcal{L}_{AN-full}$, we compare these losses on three species that are known to have interesting ranges in Figure A3.

## B ADDITIONAL DISCUSSION

### B.1 HOW DO THE BENCHMARK TASKS PROPOSED IN THIS PAPER COMPARE TO EXISTING SDM BENCHMARKS?

Presence-only SDM is notoriously tricky to evaluate (Beery et al., 2021), and there are few public benchmark datasets available for the task. Here we will discuss the two most relevant lines of prior work that have approached this evaluation problem (one from the ecology community and one from the machine learning community), and discuss where our benchmark is similar and different.

To the best of our knowledge, Elith et al. (2006) was the first attempt to systematically compare presence-only SDM algorithms across many species and locations. That work compared 16 SDM algorithms on a collection of taxonomically-specific datasets from 6 different regions, covering a total of 226 species. Presence-only data was used for training, presence-absence data was used for evaluation. Unfortunately the data was not made publicly available until Elith et al. (2020). There are two main issues with this benchmark. First, the benchmark is not suitable for studying large-scale joint SDM. It has a small number of species overall, and there are at most 54 covered in any region. Second, the species in the dataset are anonymized. This makes it impossible to use their dataset to study large-scale SDM, because we cannot increase the size of their training with external data nor can we evaluate our trained models on their test data.

Another line of work comes from the GeoLifeCLEF series of datasets and competitions (Botella et al., 2018a; 2019; Deneu et al., 2020; Cole et al., 2020; Lorieul et al., 2021; 2022). These benchmarks represent an attempt to scale up presence-only SDM, with the 2022 dataset covering 17k species with 1.6M species observations the U.S. and France. As in our benchmark, all of their training data is drawn from community science projects. The primary limitation of the GeoLifeCLEF benchmarks is that they use biased presence-only data at test time, evaluating the problem as an information retrieval task instead of a spatial prediction task.

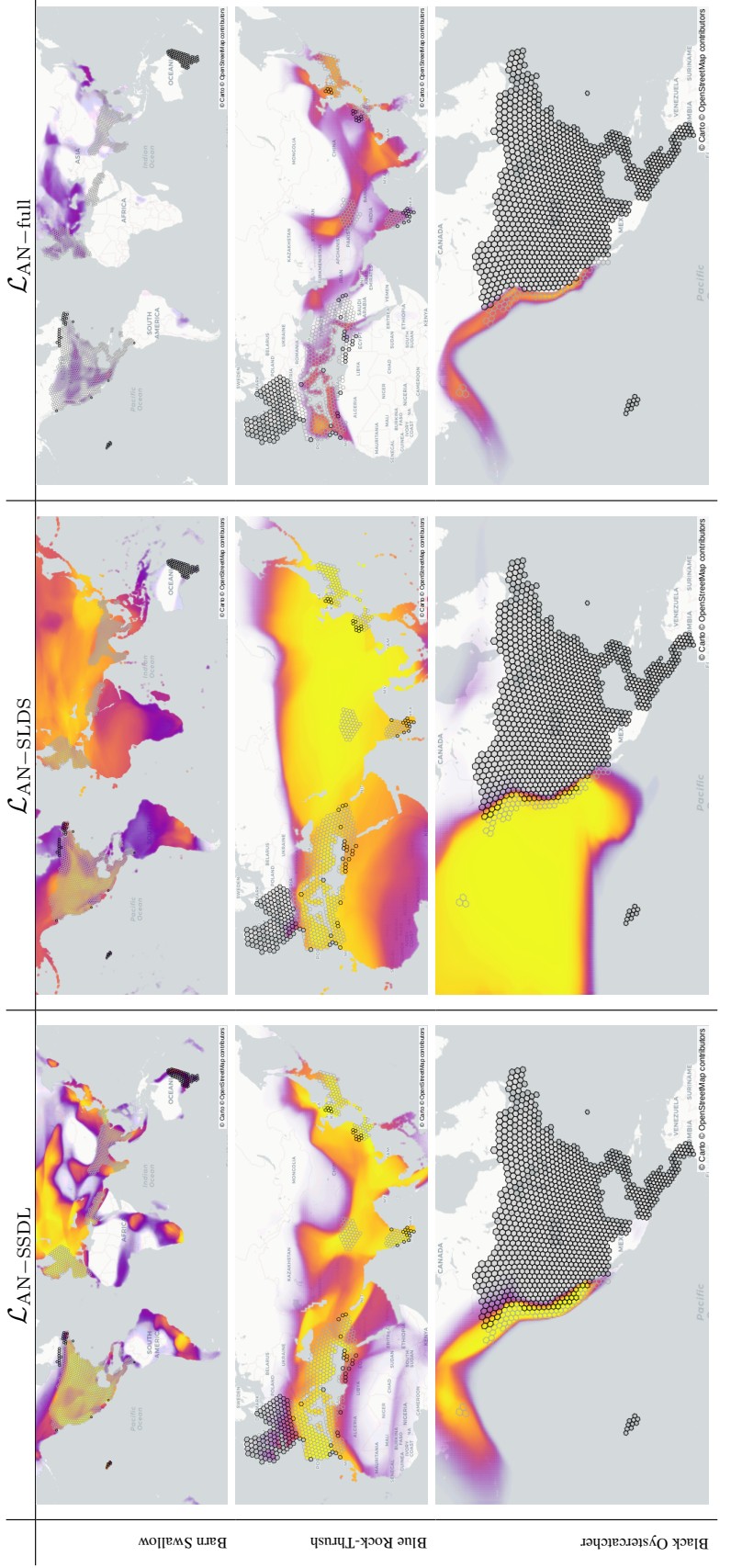

Figure A3: Loss function comparison (columns) for three different species of birds (rows). All models were trained with 1000 examples per class. See Figure 6 for an explanation of the plots. These species were chosen for visualization because their ranges have interesting complementary properties. (*Top Row*) Barn Swallow is a species that occurs across the globe. (*Middle Row*) Blue Rock-Thrush is a species whose range goes from the data rich *Western Palearctic biogeographic realm*, through a data sparse area, and back to a more data rich area of eastern Asia. (*Bottom Row*) Black Oystercatcher is a species whose range hugs the west coast of the United States. Unlike in the rest of the paper, these visualizations for Black Oystercatcher do not use a mask to filter out predictions from non-land regions. Here, we specifically wanted to see if the models learned to follow the coastline. We observe that $\mathcal{L}_{\text{AN-SLDS}}$ incorrectly expands the range into the Pacific.

Our benchmark can be viewed as a significant expansion of the GeoLifeCLEF line of work. Instead of being geographically limited to France and the U.S., we allow data from anywhere in the world. Instead of evaluating with biased presence-only data, we use presence-absence data like Elith et al. (2020). However, unlike Elith et al. (2020), we work at a large scale that allows us to study data scaling in SDM. Our indirect evaluation tasks (*Geo Prior* and *Geo Feature*) add complementary dimensions to presence-absence evaluation, and have no counterpart in Cole et al. (2020) or Elith et al. (2020).

## B.2 ENVIRONMENTAL COVARIATES VS. COORDINATES

One important characteristic of any SDM is whether or not it is *spatially explicit*. Spatially explicit SDMs include geospatial coordinates as part of the model input (Domisch et al., 2019). Traditional covariate-based SDMs include only environmental features (e.g. altitude, distance to roads, average temperature, etc.) in the input (Elith & Leathwick, 2009).

Covariate-based SDMs are often understood to reflect *habitat suitability*, because they learn a relationship between environmental characteristics and observed species occurrence patterns. A covariate-based SDM will make the same predictions for all locations with same covariates, even if those locations are on different continents. Furthermore, covariates sets must be selected by hand and are often limited in their spatial resolution and coverage.

By contrast, spatially explicit SDMs can model the fact that a species may be present in one location and absent in another, even if those two locations have similar characteristics. However, spatially explicit models are unlikely to generalize to locations that are spatially distant from the training data – such locations are simply out of distribution.

Our proposed approach is spatially explicit, so our goal is not to learn from data in one location and extrapolate to distant locations. Instead, our goal is to use abundant (but noisy and biased) species observation data to approximate high-quality expert range maps. Our locations of interest are the same during training and testing. The difference is the training data source and quality. See Merow et al. (2014) for a more nuanced discussion of extrapolation vs. interpolation and the role of model complexity in SDM.

## B.3 THE ROLE OF TIME

Some species are immobile (e.g. trees), while others (e.g. birds) may occupy different areas at different times of the year. For this reason, there has long been interest in the temporal dynamics of species distributions (Collins & Glenn, 1991; Guisan & Rahbek, 2011). However, traditional SDMs use environmental features as input, which seldom include temporal structure (Elith et al., 2020; Norberg et al., 2019). For instance, the famous WorldClim bioclimatic variables used in many SDM papers are non-temporal (Hijmans et al., 2005). It is therefore not unusual for papers on SDM to make no explicit considerations for temporal information. Similarly, in this work we do not use temporal information during training or evaluation. However, we consider this to be an interesting area for future work.

## C IMPLEMENTATION DETAILS

### C.1 NETWORK ARCHITECTURE

Unless otherwise specified, we use the network in Figure A4 for our location encoder $f_\theta$. This is identical to the architecture in Mac Aodha et al. (2019), and similar architectures have been used for other tasks (Martinez et al., 2017). The right side of the figure shows the network structure, consisting of one standard linear layer and four residual layers. The left side of the figure shows the structure of a single residual layer. Note that all layers have the same number of nodes. Every layer of the network has 256 nodes and we use $p = 0.5$ for the dropout probability.

### C.2 TRAINING DETAILS

**Environment.** All models were trained on an Amazon AWS `p3.2xlarge` instance with a Tesla V100 GPU and 60 GB RAM. The model training code was written in PyTorch (v1.7.0).

**Hyperparameters.** All models were trained for 10 epochs using a batch size of 2048. We used the Adam optimizer with an exponential learning rate decay schedule of

$$\texttt{learning\_rate} = \texttt{initial\_learning\_rate} \times \texttt{epoch}^{0.98}$$

where $\texttt{epoch} \in \{0, 1, \ldots, 9\}$. For each model we tuned the learning rate over $\{1e{-}2, 1e{-}3, 1e{-}4\}$ and selected the model with the best validation performance on the "S&T" task. For $\mathcal{L}_{\text{AN}-\text{full}}$ and $\mathcal{L}_{\text{GP}}$ we set $\lambda = 2048$.

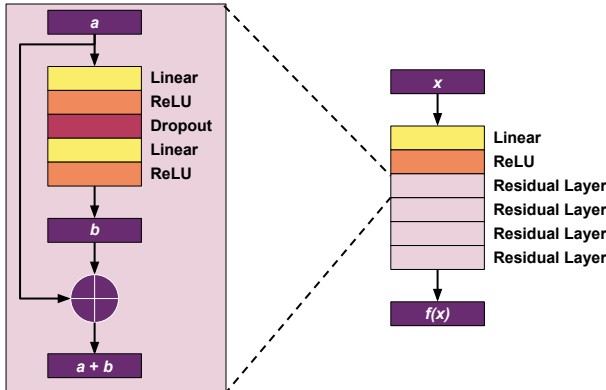

Figure A4: Network diagram for the fully connected network (with residual connections) which we use for our location encoder $f_\theta$.

### C.3 BASELINES

#### C.3.1 LOGISTIC REGRESSION WITH ENVIRONMENTAL COVARIATES

This section discusses our implementation of logistic regression with environmental covariates, whose performance is discussed in Appendix A.1.

The architecture for this approach is equivalent to using our method but replacing the location encoder $f_\theta$ with the identity function. Then we can in principle use any of our loss functions for training. All other training details follow Appendix C.2.

This approach uses environmental covariates as input, not raw coordinates. This means that environmental covariates are required for both training and evaluation. We use the elevation and bioclimatic rasters from WorldClim 2.1 (Fick & Hijmans, 2017) at the 5 arc-minute spatial resolution. We normalize each covariate independently by subtracting the mean and dividing by standard deviation (ignoring `NaN` values). We the replace `NaN` values with zeros. Note that we use these same environmental covariates as the inputs to our model in Appendix A.1.

#### C.3.2 DISCRETIZED GRID

Our baseline system for the *S&T*, *IUCN*, and *Geo Prior* tasks is a simple spatial binning method. Once we choose a resolution level, the H3 geospatial indexing library (H3W) defines a collection of $W$ hexagons $\{H_1, \ldots, H_W\}$ that partition the globe. For instance, $W = 2,016,830$ at resolution level 5. We show discretized grid results for a few different resolution choices in Table A1. Below we describe the discretized grid baseline in more detail.

For the *S&T* and *IUCN* tasks, we can compute a score for any hexagon and species as follows:

Table A1: Discretized Grid baseline results on test data when using various hexagon resolution for 'training' the model. As this baseline does not learn a location encoder, it is not possible to evaluate it on the *Geo Feature* task.

| | | Species Range | IUCN | Geo Prior | Geo Feature |
|---|---|---|---|---|---|
| Hex Res | # / Cls. | MAP | MAP | $\Delta$ Top-1 | $R^2$ |
| 0 | All | 61.50 | 94.42 | 3.5 | - |
| 1 | All | 70.05 | 90.43 | 4.1 | - |
| 2 | All | 72.87 | 81.07 | 3.1 | - |
| 3 | All | 62.85 | 69.22 | -0.9 | - |

1. We compute the number of occurrences of species $j$ in hex $w$ as

$$n_{wj} = \sum_{i=1}^{N} \mathbb{1}_{[\mathbf{x}_i \in H_w]} \mathbb{1}_{[z_{ij}=1]} \tag{5}$$

for $1 \leq w \leq W$ and $1 \leq j \leq S$.

2. Let $H_t$ be a hexagon we wish to evaluate at test time. For any species $1 \leq j \leq S$, we compute a prediction for $H_t$ as

$$\hat{y}_j = \frac{n_{tj}}{\max_{1 \leq w \leq W} n_{wj}}. \tag{6}$$

That is, $\hat{y}_j$ measures how often species $j$ was observed in $H_t$ (relative to how often species $j$ occurred in the location where it was observed most often). These predictions always fall between 0 and 1, which ensures that they are compatible with the average precision metrics we use for *S&T* and *IUCN* evaluation.

For the *Geo Prior* task, the first step is the same but the second step is different:

1. We the number of occurrences of species $j$ in hex $w$ as

$$n_{wj} = \sum_{i=1}^{N} \mathbb{1}_{[\mathbf{x}_i \in H_w]} \mathbb{1}_{[z_{ij}=1]} \tag{7}$$

for $1 \leq w \leq W$ and $1 \leq j \leq S$.

2. Let $H_t$ be a hexagon we wish to evaluate at test time. For any species $1 \leq j \leq S$, we compute a prediction for $H_t$ as

$$\hat{y}_j = \mathbb{1}_{[n_{wj}>0]}. \tag{8}$$

That is, any species which were not observed in $H_t$ are "ruled out" for the downstream image classification problem.

## D   TRAINING DATASET

### D.1   DATASET CONSTRUCTION

Our training data was collected by the users of the community science platform iNaturalist (iNa). iNaturalist users take photographs of plants and animals, which they then upload to the platform. Other users review these images and attempt to identify the species. The final species labels are decided by the consensus of the community. Each species observation consists of an image and associated metadata indicating when, where, and by whom the observation was made. iNaturalist data only contains presence observations, i.e. we do not have access to any confirmed absences in our training data.

Specifically, our training data was sourced from the iNaturalist AWS Open Dataset[2] in May 2022. We began by filtering the species observations according to the following rules:

---

[2]https://github.com/inaturalist/inaturalist-open-data

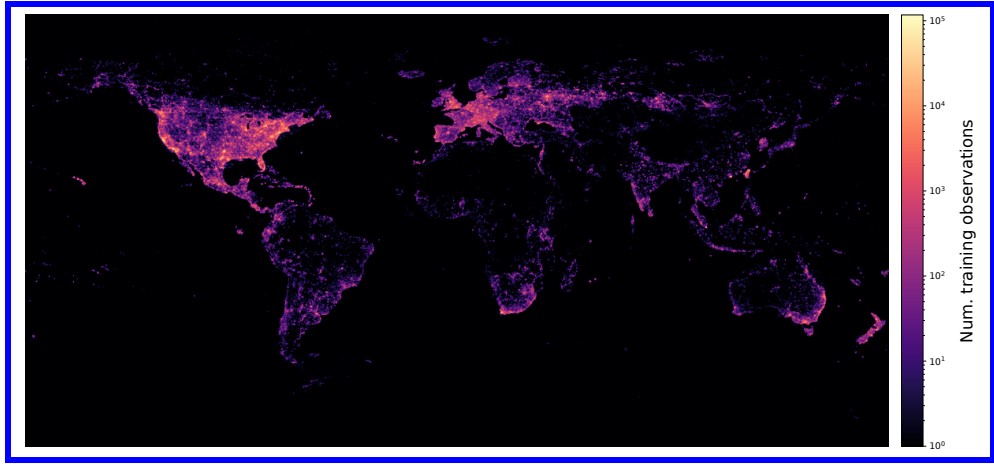

Figure A5: Histogram of the locations of the observations from our iNaturalist training set. Here we bin the data for all 35 million observations across all 47,375 species. Darker colors indicate fewer observations, brighter colors indicate more. The training data is biased towards North America, Europe, and New Zealand.

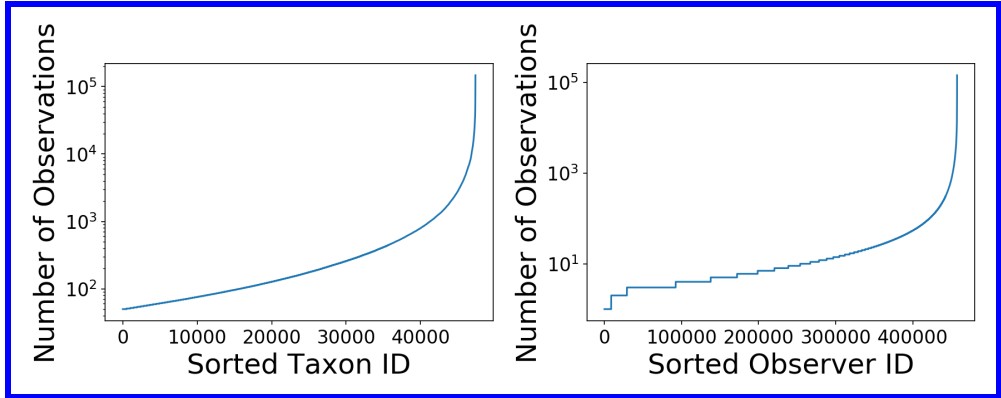

Figure A6: Summary statistics for our training data. (*Left*) Distribution of observations over species. (*Right*) Distribution of observations over users.

1. Observations must have valid valid longitude and latitude data.
2. Observations must be identified to the *species* level by the iNaturalist community. Observations which can only be identified to coarser levels of specificity are discarded.
3. Observations must have *research grade* status, which indicates that there is a consensus from the iNaturalist community regarding their taxonomic identity.

After this filtering process, species with fewer than 50 observations were removed from the dataset. We also remove any species which are marked as *inactive*[3]. Finally, we only included observations made prior to 2022. This will enable a temporal split from 2022 onward to be used as a test set in the future. After filtering, we were left with 35,500,262 valid observations from 47,375 distinct species. A visualization of the geographical distribution of the resulting data can be seen in Figure A5. As Figure A6 shows, our training data is heavily imbalanced, reflecting the natural frequency with which they are reported to iNaturalist (Van Horn et al., 2018).

## D.2 CHANGING THE NUMBER OF TRAINING EXAMPLES PER CATEGORY

In the main paper we consider the impact of the number of *observations* per species in the training set by training on different sub-sampled datasets. We construct these datasets by choosing $k$ obser-

---

[3]https://www.inaturalist.org/pages/how+taxon+changes+work

vations per species, uniformly at random. We set a seed to ensure that we are always using the same $k$ observations per category. We also make certain that sampled datasets are nested, so the dataset with $k_1$ examples per category is a superset of the dataset with $k_2 < k_1$ examples per category. If a category has fewer than $k$ observations, we use them all.

### D.3  CHANGING THE NUMBER OF TRAINING CATEGORIES

In the main paper we also consider the impact of the number of *species* in the training set. In particular, we consider the following nested subsets:

- The set of 536 bird species in the eBird Status & Trends dataset (Fink et al., 2020).

- The eBird Status & Trends species plus an additional $A$ randomly selected species, where $A \in \{4000, 8000, 24000\}$.

## E  EVALUATION TASKS

Here we provide additional details on the benchmark tasks used in the main paper. For each task, we outline the dataset properties, how it was collected, and the evaluation metrics used.

### E.1  S&T: EBIRD STATS AND TRENDS

**Task:** The goal of this task is to evaluate how effective models trained on noisy crowdsourced data from iNaturalist are at predicting species range maps. We use the *eBird Status & Trends* data from Fink et al. (2020) to evaluate our range predictions. This dataset consists of estimated range maps for 536 species of birds predominantly found in North America, but also other regions. The range maps are the output of an expert crafted model (Fink et al., 2020) that has been trained on tens of millions of presence and absence observations, makes use of additional expert knowledge to perform data filtering, and uses rich environmental covariates as input. While not without its own limitations, we treat this data as 'ground truth' for evaluation purposes as it is developed using much higher quality data compared to what we use to train our models.

**Dataset:** We perform several pre-processing steps on the raw data. We first spatially bin the data using hexagons at resolution five as defined by the H3 project [4]. $2,016,842$ hexagons cover the world at this resolution, each with an average area of $252.9\text{km}^2$. Our goal is to predict the presence or absence of a given species in each hexagon using the *eBird Status & Trends* output as 'psuedo' ground truth. The evaluation regions are restricted to those where the *eBird Status & Trends* models have determined that there is sufficient data to make a prediction for a given species. Thus, the set of evaluation regions can vary from species to species. For example, `Melozone aberti` has 23,282 locations with known presence or absence, of which 271 are deemed present. On the other hand, `Columba livia` has 67,251 locations with known presence or absence, of which 39,754 are deemed present.

The *eBird Status & Trends* data provides species presence and absence information for each location over the course of the year. For the purposes of our evaluation, we collapse all time points to one, and count a hexagon region as being a presence for a given bird if the output of their model is greater than zero for any week in the year for that species.

We create a validation split by choosing 50% of the hexes with known presence/absence status uniformly at random for each species. This is in line with the notion of generalization we care about. That is, we want to know how well we can reproduce expert range maps from raw, potentially noisy, data, *not* how well we can make predictions in previously unseen locations.

**Evaluation:** We use mean average precision (MAP) as our evaluation metric, only evaluating on valid regions for a given species.

---

[4]https://h3geo.org

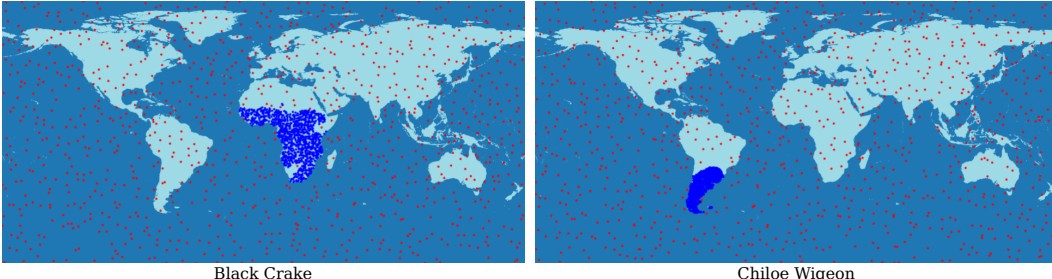

Black Crake          Chiloe Wigeon

Figure A7: Here we show two example species from the IUNC range map evaluation task. The blue points represent positive observations, i.e. where the expert generated range maps indicate that the species is likely to be found. In contrast, the red points indicate absences.

### E.2 ICUN: RANGE MAPS

**Task:** The goal of this task is similar to the previous one, i.e. to predict the geographical range of a set of species. However, instead of 'ground truth' target maps that are estimated by another model, here we use expert curated range maps from the International Union for Conservation of Nature (IUCN) [5]. This set of data contains a more taxonomically diverse set of species compared to the *Stats and Trends* task, as it contains mammal and fish species, among others, in addition to birds.

**Dataset:** We select a subset of 3,938 species that overlap with our training species. For each species, we take the expert generated range polygon(s), which indicates the geographical range of a given species, and sample a set of points inside the polygons(s) and denote them as *presence* observations. In addition, we sample another set of points, uniformly at random outside those polygons(s) and denote them as *absence* observations. In total, we sample 2,000 observations per species, half presence and half absence. A visualization of the resulting sampling process is shown in Figure A7.

Note that the absolute performance on this dataset in Table 1 is often well above 90%. We believe this is due to the fact that the absence samples are "easy" because many of them are far away from the presence samples. Alternative sampling processes could have been used, e.g. with more emphasis on the boundary regions between presence and absence areas as denoted by the expert maps. However, as the expert maps cannot necessarily be assumed to be the objective ground truth (i.e. species ranges can shift over time), we instead opted for the strategy outlined above.

**Evaluation:** Again we use mean average precision (MAP) as the evaluation metric, which results in a single score for a model averaged across all species.

### E.3 GEO PRIOR: GEOGRAPHICAL PRIORS FOR COMPUTER VISION

**Task:** The goal of this task is to combine the outputs of the models trained for species range estimation on the iNaturalist dataset with computer vision image classifier predictions. This evaluation protocol has also been explored in other work, e.g. Berg et al. (2014); Mac Aodha et al. (2019). We simply weight the probabilistic image classifier predictions for a given image with the species occupancy predictions from the location where that image was taken. The intuition is that the occupancy prediction down-weights the probability of a given species being predicted by the vision model if the range estimation model predicts that the species is *not* likely to be present at that location.

**Dataset:** For the vision classifier, we use an image classification model developed by the iNaturalist community science platform [6]. This model is an Xception network (Chollet, 2017) that has been trained on 55,000 different taxonomic entities (i.e. classes) from over 27 million images. We take the predictions from the final classification layer of the classifier, and do not apply any of their sophisticated taxonomic post-processing. There are a total of 49,333 species in the set of 55,000 classes, the rest are higher order groups, e.g. genera. The images used to train the image classifier come from observations that were added to iNaturalist prior to December 2021.

---

[5]https://www.iucnredlist.org/resources/spatial-data-download
[6]https://www.inaturalist.org/blog/63931-the-latest-computer-vision-model-updates

We then constructed a test set consisting of all research grade observations (i.e. those observations for which there is a consensus from the iNaturalist community as to which species is present in the image). The images in the test set only contain the set of species that were observed at training time, i.e. we do not consider the open-set prediction problem. The observations were selected from between January and May 2022 to ensure that they did not overlap with the training set. We take at most ten observations per species, which results in 282,974 total observations from 39,444 species. In practice, many species do not have 10 observations. In total there are 2,721 species, with 9,808 total images, that are not present in our range estimation training set. For each of the 282,974 observations, we extract the predictions from the deep image classifier across all 39,444 remaining species.

**Evaluation:** Performance is evaluated in terms of top-1 accuracy, where the ground truth species label is provided by the iNaturalist community. Without using any information about where an image was taken, the computer vision model alone obtains an accuracy of 75.4%, which increases to 90.4% for top-5 accuracy. During evaluation, if a species is not present in a range model, we simply set the output for the range model for that species to 1.0.

### E.4 Geo Feature: Environmental Representation Learning

**Task:** This task aims to evaluate how well features extracted from deep models trained to perform species range estimation can generalize to *other* dense spatial prediction tasks. Unlike the other benchmark tasks that use the species occupancy outputs directly, this is a transfer learning task. We remove the classification head $h_\phi$ and evaluate the trained location encoder $f_\theta$ in terms of downstream environmental prediction tasks. The intuition is that a model that is effective at range estimation may have learned a good representation of the local environment. If so, that representation should be transferable to other environmental tasks with minimal adaptation.

This task is inspired by the linear evaluation protocol that is commonly used in self-supervised learning, e.g. Chen et al. (2020). In that setting, the features of the backbone model are frozen and a linear classifier is trained on them to evaluate how effective they are on various downstream classification tasks. In our case, instead of classification, we aim to *regress* various continuous environmental properties from the learned feature representations of our range estimation models. A related evaluation protocol was recently used in Rolf et al. (2021) for the case of evaluating models trained on remote sensing data.

**Dataset:** The task contains nine different environmental data layers which have been collected using Google Earth Engine [7]. The nine data layers are described in Table A2. For each of the layers, we have rasterized the data so that the entire globe is represented as a $2004 \times 4008$ pixel image. Each pixel represents the measured value for a given layer for the geographical region encompassed by the pixel. Example images can be seen in Figure A8.

**Evaluation:** For evaluation, we crop the region of interest to the contiguous United States and grid it into training and test cells. The spatial resolution of the training and testing cells are illustrated in Figure A8 (right). Note, we simply ignore locations that are not in the training or test sets, e.g. the ocean. This results in 51,422 training points and 50,140 test points. Features are then extracted from the location encoder for the spatial coordinates specified in the training split, and then a linear ridge regressor is trained on the train pixels and evaluated on the held out test pixels. The input features are normalized to the range $[0, 1]$. We cross validate the regularization weighting term $\alpha$ of the regressor on the training set, exploring the set $\alpha \in \{0.0001, 0.001, 0.01, 0.1, 1.0, 10.0\}$. Performance is reported as the coefficient of determination $R^2$ on the test pixels, averaged across all nine layers.

## F Reproducibility Statement

The information needed to implement and train the models outlined in this paper are provided in Section C. In addition, the different training losses evaluated are described in Section 3.2. Upon acceptance, we will provide code to enable researchers to build on our results.

---

[7]https://earthengine.google.com

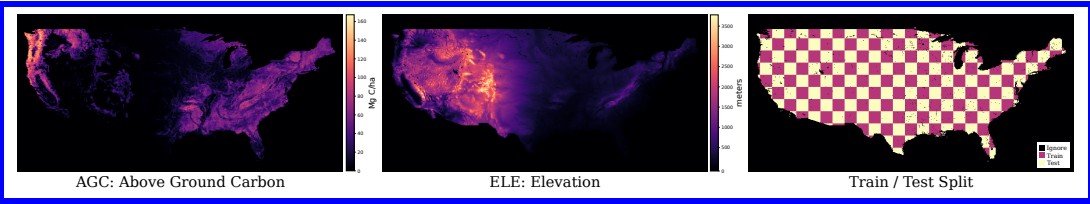

| AGC: Above Ground Carbon | ELE: Elevation | Train / Test Split |

Figure A8: Here we illustrate two of the nine evaluation layers used in the *Geo Feature* prediction task (left and middle). On the right we indicate which regions contain pixels that are in the train or test split, or simply ignored during evaluation.

Table A2: Description and sources of the nine environmental spatial layers that are part of our *Geo Feature* prediction task.

| Name | Task | Units | Range |
|------|------|-------|-------|
| AGC | Above ground living biomass carbon stock density of combined woody and herbaceous cover in 2010. `NASA/ORNL/biomass_carbon_density/v1 - agb` | Mg C/ha | 0 to 129 |
| ELE | GMTED2010: Global multi-resolution terrain elevation data 2010. Masked to land only. `USGS/GMTED2010 - be75` | meters | -457 to 8746 |
| LAI | The sum of the one-sided green leaf area per unit ground area. `JAXA/GCOM-C/L3/LAND/LAI/V2 - LAI_AVE - 2020` | (see previous) | 0 to 65531 |
| NTV | Percent of a pixel which is covered by non-tree vegetation. `JAXA/GCOM-C/L3/LAND/LAI/V2 - LAI_AVE - 20202` | % | 0 to 100 |
| NOV | Percent of a pixel which is not vegetated. `MODIS/006/MOD44B - Percent_NonVegetated` | % | 0 to 100 |
| POD | UN adjusted estimated population density. `CIESIN/GPWv411/GPW_UNWPP-Adjusted _Population_Density - unwpp-adjusted_population_density` | # of persons / km$^2$ | 0 to 778120 |
| SNC | Normalized difference snow index snow cover. `MODIS/006/MOD10A1 - NDSI_Snow_Cover - 2019` | (see previous) | 0 to 100 |
| SOM | Soil moisture, derived using a one-dimensional soil water balance model. `IDAHO_EPSCORTERRACLIMATE - soil - 2020` | mm | 0 to 8882 |
| TRC | The percentage of pixel area covered by trees. `NASA/MEASURES/GFCC/TC/v3 - tree_canopy_cover - 2000-2020` | % | 0 to 100 |

## G  ETHICS

This work makes use of species observation data provided by the iNaturalist community. We explicitly only use data from the public data exports from iNaturalist to ensure that no data, that is not already available publicly, related to sensitive species (e.g. those who are at risk of extinction) is used by our models. As noted in the limitations section in the main paper, extreme care needs to be taken when attempting to interpret any species range predictions from the models presented in this paper. The models have not been evaluated on those types of downstream tasks.

