# OpenReview forum: "Global-Scale Species Mapping From Crowdsourced Data"
_ICLR.cc/2023/Conference — Submitted to ICLR 2023_

### Official Review · Reviewer_eJKM · 2022-10-24

**Confidence:** 4
**Correctness:** 3
**Technical Novelty And Significance:** 2
**Empirical Novelty And Significance:** 2
**Recommendation:** 3

**Clarity, Quality, Novelty And Reproducibility:**

This study is intuitively and technically sound. In addition, the paper is presented clearly and is easy to follow and reproduce. The experimental design is adequate, and the method is simple but the novelty is limited.

**Strength And Weaknesses:**

This study is intuitively and technically sound. In addition, the paper is presented clearly and is easy to follow and reproduce. The main idea of the proposed method is simple but effective.

However, there are several weaknesses:
- Overall, the novelty of this paper is limited. The method proposed in this paper seems a combination of existing works, which mainly builds on the work of Mac Aodha et al. (2019) and Cole et al. (2021). Furthermore, the discussion of the main idea and the structure of the proposed method is too brief. I suggest the author discuss the main contributions of this paper and highlight the innovation compared with existing works, and describe the proposed framework more structurally.
- The analysis of the experimental results is somewhat weak. It is mentioned in Section 2 that the authors will attempt to answer the question about what training loss is most effective, but in the section of experiments, this topic has not been discussed in depth, since no loss is best across all tasks according to the experimental results, it’s better to discuss about why this happened, and analyze which types of tasks the different loss are applicable to? Furthermore, the authors conduct several variants to highlight the importance of the deeper location encoder in terms of learning an effective representation of space, however, the proposed method is only compared with the “single layer” case, and it is not convincing enough to support the claim, and why the authors choose the network with four residual layers for location encoder?


**Summary Of The Paper:**

This paper proposes a novel method that estimates the geographical ranges of different species based on presence-only data. This paper combines the idea of learning multi-label classifiers from limited supervision with implicit coordinate networks to improve the performance of species range estimation. The authors conduct a series of experiments to show the effectiveness of the proposed method.

**Summary Of The Review:**

Overall, the paper has its merits to show a new method for species range estimate, and improve the performance of the model. However, due to some key issues that have not been well addressed, this paper can hardly meet the high-quality requirements of ICLR.

---

> ### Author Response · Authors · 2022-11-15
> **Response for eJKM**
>
> # E1. Machine learning novelty.
>
> We address this comment in the general response (see C1).
>
> # E2. Discussion of the main idea is too brief.
>
> We have expanded the explanation of the explored losses by highlighting the different forms of supervision and assumptions they make **in our new Figure 2**.
>
> # E3. Relationship to Cole et al. 2021 and Mac Aodha et al. 2019.
>
> Cole et al. outline a new multi-label learning setting called single positive multi-label learning (SPML). Our species range mapping task is an instance of SPML. However, the approach proposed in Cole et al. (ROLE) requires the estimation of a large label matrix (number of observations times the number of categories). Given the large size of our dataset, it is not feasible to run ROLE due to memory limitations. However, we compare against a recent state-of-the art SPML method from Zhou et al. 2022 which beats the method of Cole et al. on multi-label image classification. We observe in Table 1 ($\mathcal{L}_\mathrm{ME-X}$ rows), that their method does not perform well on most tasks, with the exception of the *Geo Feature* task. This indicates that our benchmarks tasks will be of particular interest to the SPML community.
>
> As noted in Section 3.2, our $\mathcal{L}_\mathrm{AN-full}$ loss is a simplification of the loss proposed in Mac Aodha et al. One can view $\mathcal{L}_\mathrm{AN-SSDL}$ and $\mathcal{L}_\mathrm{AN-SLDS}$ as ablations of $\mathcal{L}_\mathrm{AN-full}$ that allow us to understand the relative importance of different kinds of pseudo-negatives, which was not explored in Mac Aodha et al. In addition, Mac Aodha et al. only evaluate on the *Geo Prior* task. We evaluate the *Geo Prior* task at an unprecedented scale, and we introduce new geospatial tasks that are complementary.
>
> # E4. Analysis of the experimental results is weak.
>
> We address this comment in the general response (see C3).
>
> # E5. Choice of network architecture.
>
> We address this comment in the general response (see C4).

---

> ### Author Response · Authors · 2022-11-28
> **Please let us know if you have any further questions!**
>
> We have provided comments that we believe address your concerns, but please let us know if there is anything else we can clarify!

---

> ### Author Response · Authors · 2022-12-12
> **Happy to answer any unresolved questions!**
>
> The review period ends today, but we are still eager to respond to any outstanding questions or concerns!

---

### Official Review · Reviewer_f3dW · 2022-10-24

**Confidence:** 4
**Correctness:** 2
**Technical Novelty And Significance:** 2
**Empirical Novelty And Significance:** 3
**Recommendation:** 3

**Clarity, Quality, Novelty And Reproducibility:**

The presentation of the paper, especially the Methods section, is not clear and needs significant improvement.
This is a good application-oriented paper, which adopts some existing frameworks and methods, but there is no novel method proposed.
The author has provided the links to the dataset used in the experiment, and, as the author claimed, "Up on acceptance, we will provide code to enable researchers to build on our results."


**Strength And Weaknesses:**

Strength:
The paper works on a significant application in ecological domain: species range estimation at a global scale from large-scale, presence-only, crowdsourced data.
The proposed 4 benchmark evaluation tasks and the corresponding datasets may encourage and help domain researchers who want to work on species distribution problem.

Weakness:
Most citations in the article do not meet the format requirements.
The presentation of the paper needs improvement. Firstly, there are some typos and grammar errors in this paper, e.g., line 4 of Section 1: to understanding -> to understand, line 2 of the 2nd paragraph of Section 2: decision tress -> decision trees, line 4 of Section 4.6: explicitly attempted -> explicitly attempt. Secondly, the description of the method is hard to understand. The paper needs to add illustrations for the proposed loss function, highlight the innovative contribution, explain and justify these loss functions in detail.
The technical novelty of this paper is not sufficient. The method in this paper only adopts the framework in "Elijah Cole, Oisin Mac Aodha, Titouan Lorieul, Pietro Perona, Dan Morris, and Nebojsa Jojic. Multi-label learning from single positive labels. In CVPR, 2021." to define the loss functions, and use some classic neural network architecture to train the vector representation of space.
The experiment result is not convincing since the paper only evaluates the performance of the methods proposed in this paper and their variants on some self-collected datasets. The author needs to make comparisons to state of the art alternative methods and test the effectiveness of the proposed method on some widely recognized benchmarks in the species distribution modeling research domain.
The analysis of the experiment result is insufficient. The author only demonstrates the result, e.g., the “best” loss depends on the task, and method rankings may not generalize across domains. The paper does not explain these results in detail.


**Summary Of The Paper:**

The paper proposes a method to handle the species distribution modeling (SDM) problem at a global scale, which is challenging and important. Traditional models need presence-absence data for training and are unable to take advantage of large-scale data. In contrast, with several scalable loss functions, the method in this paper can use large-scale presence-only data for the SDM problem. The paper also proposes 4 benchmark evaluation tasks to measure different aspects of the SDM problem and presents the experimental result conducted on these tasks to test the effectiveness of the proposed method.


**Summary Of The Review:**

This paper addresses an important problem, i.e., species range estimation at a global scale from large-scale, presence-only, crowdsourced data, and provides 4 new benchmark tasks. However, the presentation of this paper needs improvement and the technique novelty of this paper is not sufficient. Also, this article lacks sound experiments and reasonable analysis. Due to these reasons, we recommend rejecting the paper.

---

> ### Author Response · Authors · 2022-11-15
> **Response for f3dW**
>
> # F1. Reference formatting is incorrect.
>
> Thanks for pointing this out! **We have fixed this issue in the revised text.**
>
> # F2. Typos.
>
> **We have addressed the typos you flagged in the revised text.**
>
> # F3. Clarify the description of the methods.
>
> **We have added a new figure (Figure 2) that illustrates the different sources of supervision used by the different loss functions.**
>
> # F4. Only report results on variants of the proposed losses.
>
> We have responded to this question in the general response to all reviewers (see C2). As noted above, we also compare to several existing methods in the initial draft.**As requested by reviewers, we have also added results for new comparisons to the literature in the form of models that use additional environmental features as input during training and inference. Our approaches (which use only coordinates as input) outperform the traditional methods with environmental covariates (logistic regression). When we use environmental covariates as an input to our method, we gain only +1.1 MAP (compared to logistic regression which gains over +15 MAP), indicating that we capture much of the value of environmental covariates without training on them explicitly.**
>
> # F5. Report results on widely recognized benchmarks.
>
> **To the best of our knowledge there are no large-scale widely recognized benchmarks in the species distribution modeling research domain.** That points to the importance of one of our main contributions. One of the most widely cited evaluation datasets from the ecology literature for this task only contains data for 225 species, localized in small geographical regions (see Elith et al. Ecography 2006). In addition, the species are anonymized and the raw location information for half of these species has been obfuscated, so it is impossible to evaluate our trained models on their benchmarks. We have added a discussion of these issues in Appendix B.1. If you are aware of any other datasets of similar scale and quality to ours please let us know.
>
> # F6. Missing analysis of the results.
>
> We have addressed this question in the general response to all reviewers (see C3).
>
> # F7. Presentation of this paper needs improvement.
>
> We have made significant edits for text quality and clarity based on the helpful comments of the reviewers. If you see any specific areas for improvement in the updated version, please let us know.

---

> ### Author Response · Authors · 2022-11-28
> **Please let us know if you have any further questions!**
>
> We have provided comments that we believe address your concerns, but please let us know if there is anything else we can clarify!

---

> ### Author Response · Authors · 2022-12-12
> **Happy to answer any unresolved questions!**
>
> The review period ends today, but we are still eager to respond to any outstanding questions or concerns!

---

### Official Review · Reviewer_apN4 · 2022-10-25

**Confidence:** 4
**Clarity, Quality, Novelty And Reproducibility:** 1.	Overall, this article proposes thr…
**Correctness:** 3
**Technical Novelty And Significance:** 2
**Empirical Novelty And Significance:** 2
**Recommendation:** 3

**Strength And Weaknesses:**

Strength:
1.	Clearly specify the background and problem definition.
2.	Give a clear description of the proposed methods (three different loss functions).
3.	The author created four different evaluation tasks to justify the effectiveness of the proposed model. The experiment analysis is very clear and easy to understand. In particular, the author shows that the three proposed loss functions have different performances in different tasks.
Weaknesses:
1.	Although the authors mentioned the proposed loss functions in the methodology section, it did not show the overall architecture of the proposed model. Which makes readers not fully understand the proposed methods.
2.	Although the author shows great experiment results and analysis in the evaluation section, the author didn’t compare their proposed method with state-of-the-art methods(ML methods and statistic methods). Thus, we don’t know whether the proposed methods outperform other methods or not.


**Summary Of The Paper:**

In this paper, the authors proposed a new approach(with 3 different loss functions) to jointly estimate the geographical ranges of tens of thousands of different species simultaneously. At the same time, the authors develop a series of experiments under different settings to justify the effectiveness and robustness of the proposed methods.

**Summary Of The Review:**

Overall, this article proposes a new approach to achieve the goal of estimating the geographical ranges of species simultaneously, but the method lacks novelty. The experiment results from four different can justify the effectiveness of the proposed method, but the authors didn’t compare their methods with other state-of-the-art methods (ML methods and statists methods).

---

> ### Author Response · Authors · 2022-11-15
> **Response for apN4**
>
> # A1. Did not show the architecture of the proposed model.
>
> We illustrated and described the network architecture used in our experiments in Figure A.1 and Section A.1 of the submitted paper. **We have amended the text in Section 4.1 of the main paper to make it clear to the reader that the model architecture can be found there. We have also added a new figure (Figure 2) that illustrates the different sources of supervision used by the different loss functions.** If any aspect of the architecture still remains unclear, please let us know.
>
> # A2. Lack of comparisons to existing methods.
>
> We have addressed this question in the general response to all reviewers (see C2).
>
> # A3. Lack of novelty.
>
> We have addressed this question in the general response to all reviewers (see C1).

---

> ### Author Response · Authors · 2022-11-28
> **Please let us know if you have any further questions!**
>
> We have provided comments that we believe address your concerns, but please let us know if there is anything else we can clarify!

---

> ### Author Response · Authors · 2022-12-12
> **Happy to answer any unresolved questions!**
>
> The review period ends today, but we are still eager to respond to any outstanding questions or concerns!

---

### Official Review · Reviewer_Jszv · 2022-10-27

**Confidence:** 3
**Correctness:** 2
**Technical Novelty And Significance:** 2
**Empirical Novelty And Significance:** 3
**Recommendation:** 6

**Clarity, Quality, Novelty And Reproducibility:**

**Clarity**: The paper is mostly well written and easy to follow. There are some mistakes/typos that make some sentences odd, but it should be easily fixable with one careful proof-read.

**Quality**: The method looks sound and benchmarking tasks look reasonable. The empirical evaluation as mentioned above could be better as the proposed approach is compared with only one existing work (in addition to some baselines). Why certain choices were made is not clear: for example, why this particular architecture for a neural network was chosen.

**Novelty**: The use of location only information appears to be novel, but ideas about different ways for dealing with the presence-only issue have been explored before (please note the paper doesn’t claim the opposite). The benchmarking tasks appears novel and are as it seems as big contribution as the idea of using the location only data. However, this novelty comes from the application perspective. There is not too much pure machine learning novelty.

**Reproductivity**: The paper seems to provide the main information for reproducibility. Some details would be better to add, such as more explicit how the mean average precision was calculated

Detailed comments/suggestions (mostly not affecting the score):
1. Section 2. “A subset of these methods…” – unfinished sentence
2. Some conventions are not typical for machine learning literature:
    * reference style with Author (Year) even for references that appear at the end of the sentences;
    * separation of results and discussions
3. After eq. (2). r\sim Unif(X). Unif is not defined
4. Discussion on not using time and how this fits with the research question and how to prepare data for this could be transferred in the main text as it did raise questions for me before I read the appendix. Actually it still sometimes unclear for the settings. Is the goal to extrapolate the information about presence of species to different locations? Or is the goal to predict presence of species at a different from training time?
5. Section 4.3. Geo Feature. “features for a dense set of spatial locations” is unclear
6. S&T data. Does the time of the data overlaps with the training data? Is it ok if it does?
7. Table 1. Geo Prior column. Is it in comparison to the model trained on images only? This should be specified
8. Missing legend for Figures 4 and A2 and A4
9. “Few exceptions” in section 4.5 “Joint learning…” – would be great to have any possible explanation on them
10. Figure 6, first column. Hexagons are not visible unless a lot of zoom is involved. It is better to move it in the appendix and make it bigger
11. Figure 6, caption. Odd sentence “Black Oystercatcher is a coastal species the hugs the west coast of the United States”
12. Section A.3. Missing reference for H3 library
13. Section A.3 is unclear for me how these baselines work
14. Section B.3. First bullet point – missing reference
15. Section C.1 Odd sentence “eBird Status & Trends predictions for a given species are made a multiple different time points…”
16. Section C.4. Evaluation. Individual layers would also be interesting to see. Also I am curious to see performance of a baseline: a ridge regressor trained on (lat, lon) data directly. That would show the benefit of the learnt features.

Minor:
1. Section 2. Missing blank between “encodings” and “Tancik et al. (2020)”
2. Section 2. Missing year in the Yang et al. (2022, ?)
3. Section 4.4, first paragraph. L_{SLDS} -> L_{ME_SLDS}?
4. Section C.1. “This is in line we” -> “This is in line with”?
5. Section C.3. “The observations where selected” -> “The observations were selected”?


**Strength And Weaknesses:**

**Strengths**:
* The approach to use location information only appears to be novel which is surprising considering the previous approaches utilising much richer data for the same task
* Benchmarking tasks illustrating how the location data can be used
* Exploration of different strategies for pseudo absence labels

**Weaknesses**:
* Lack of comparison with the existing methods. I appreciate it would be unfair comparison as the other methods use more information, e.g. environmental data, but it would put this approach in the context of the existing literature
* [Potential]. May be not in the scope for ICLR (see details below)
* Crowdsourced data in the title is a bit misleading. There is nothing on normal crowdsourcing tasks in the paper. ”Crowdsourced data” here only refers as not too reliable and not too costly.


**Summary Of The Paper:**

The paper proposes an approach for species distribution modelling based on using location information only and presence-only information. To deal with presence-only the paper explores different loss functions how one may add pseudo absence labels. The paper also proposes a set of benchmarking tasks.

**Summary Of The Review:**

I honestly do not know whether this kind of work is in scope of ICLR. I completely accept that it may be just my bias but I haven’t seen such application papers before. There is no novelty on the actual machine learning side. The novelty comes from using the new input (location only) and presenting the new benchmarking tasks.

Aside from this, the only major issue with the paper is that there is no much comparison with the existing methods.

Giving benefit of the doubt regarding the fitness of the topic to ICLR, I am leaning towards acceptance.

*Question for the rebuttal*: Can the authors explain why other existing methods were not used in comparison? Am I missing something and they are incomparable beyond the fact that they would use more information than the proposing approach?

---

> ### Author Response · Authors · 2022-11-15
> **Response for jszv**
>
> # J1. Compared with only one existing work.
>
> Apologies if we failed to make this clear in the text. We do indeed compare to several variants of our method, but we also compare to several methods that are our implementations of existing work. These comparisons include Zhou et al. 2022, Mac Aodha et al. 2019, and Berg et al. 2014. **We have revised the text in Table 1 to make it clear which existing papers the different models correspond to.** We also provide further discussion and new results regarding comparisons to the existing literature in our general comments to all reviewers (see C2).
>
> # J2. Not in scope for ICLR.
>
> We address this concern in our general comments to reviewers (see C1). In summary, we identify several recent works that are similar in “flavor” in that their main contributions are centered on new datasets and model evaluation.
>
> # J3. The use of the term “crowdsourcing” in the paper title.
>
> We used the term “crowdsourcing” in the paper title as the data used for model training was crowdsourced from citizen scientists. However, it is true that our paper is not related to novel methods for resolving ambiguity in crowdsourced labels (as in e.g. Welinder et al. NeurIPS 2010). **To reduce any future ambiguity or misinterpretation we have changed the title of the paper to “Joint Implicit Neural Representations for Global-Scale Species Mapping“.**
>
> # J4. Typos and missing legends in figures.
>
> Thanks for flagging these! **We have corrected them in the revised text.** Note that we have not added a color bar to the ICA plot because the colors only indicate relative similarity within an image.

---

> > ### Comment · Reviewer_Jszv · 2022-11-18
> > **Thank you for your responses**
> >
> > Thank you for your responses.
> >
> > J1 - cleared, thank you
> >
> > J2 - After the second thoughts and after reading your reply, I believe the paper should be restructured to reflect what you are saying and made it more in scope of ICLR. You are saying that one of your main contributions is: "(iii) providing a new benchmark suite of tasks for evaluating large-scale range mapping and environmental representation learning", but the new benchmark tasks appear for the first time in the experiment section of the paper. For example, in the papers you mentioned where the main contribution is a dataset, its' description would be in the place where a method description would be in a paper where the main contribution is a method.
> >
> > Therefore, I believe the paper should be presented in the way of reflecting the main contributions listed by the authors in their response, i.e. it would be presenting the benchmark tasks and losses for presence-only data and then the method on location only data as a new baseline for these new benchmark datasets. Now, the paper structure suggests that the main novelty is the method which is a "standard" (i.e. not novel) neural network with the location only data. Then discussion of losses for presence-only data which is not novel. And then it is demonstrated on the new benchmark tasks. The main contribution does not appear in the experiment section normally.
> >
> > I believe this is the main reason for the current questions from all reviewers about ML novelty. If the paper is indeed structured in the way so the new benchmark tasks are put in the central place as the main contribution, then it would be easier to see its' fit at ICLR.
> >
> > Just a note regarding the papers provided as examples where datasets are the main contributions, those papers present more generic datasets for ML. For example, from Sagawa et al. "These datasets span a wide range of applications (from histology to wildlife conservation), tasks (classification, regression, and detection), and modalities (photos, satellite images, microscope slides, text, molecular graphs). " Here it would be one application, several tasks and one modality (or two if to contrast location-only and environmental features). However, this is just a note, this is not to say that the provided benchmark tasks would not be enough, but it would require a significant revision of the paper.
> >
> > J3-J4. Thank you for incorporating my suggestions

---

> > > ### Author Response · Authors · 2022-11-22
> > > **Thank you for the suggestion!**
> > >
> > > That's a good idea, thank you! **We will move Section 4.2 ("Training Data") and Section 4.3 ("Evaluation Tasks and Metrics") to the methods section** to emphasize the novelty and significance of the dataset contribution. **We will also change the ordering of the contributions** at the start of page 2 to make the dataset contribution more prominent.
> > >
> > > These standardized benchmark datasets are meant to enable future researchers to make progress on important geospatial representation learning tasks. We also contribute detailed empirical results for a diverse collection of methods, most of which have never been studied at this scale before. Taken together, we believe that these benchmarks and comparisons are relevant to ICLR.

---

### Author Response · Authors · 2022-11-15
**General Comments**

## We have uploaded a new version of the paper. New or substantially revised text appears in blue. New or updated figures are framed by a blue box. Unless otherwise specified, references (e.g. tables/figures/sections) are with respect to the updated version of the paper.

---

> ### Author Response · Authors · 2022-11-15
> **General Comments (2/4)**
>
> # C2. Comparisons to existing methods [Jszv, f3dW, eJKM]
>
> **Many existing methods for SDM do not scale to the large number of classes and training examples explored in our work. We therefore chose representatives from the limited number of applicable families of methods from the existing literature.** We trained and evaluated these methods under equivalent settings to ensure fair comparisons. Evaluated methods included a state-of-the-art single positive multi-label learning method (Zhou et al. 2022), a related deep learning method that uses additional side information (Mac Aodha et al. 2019), and a non parametric data discretization approach (Berg et al. 2014). This is in addition to other baselines e.g. logistic regression (Pearce & Ferrier 2000). We show that even recent approaches that have been designed for single positive multi-label learning (i.e. Zhou et al. 2022) do not outform our proposed approach. **Note that we do not claim that our approach is technically complex - to the contrary, we believe its simplicity is a significant strength.**
>
> We agree that it is interesting to ask how well models that use additional environmental features as input perform. **To explore this, we compare our method against the classic and scalable logistic regression approach to SDM (Pearce & Ferrier 2000). Both our method and logistic regression are evaluated using coordinate inputs and environmental feature inputs.** The new results can be found in Table 1 and Figure A1. We trained these models using the commonly used bioclimatic features from WorldClim 2.1 (https://www.worldclim.org/data/bioclim.html). From the results (Table 1, Figure A1, and reproduced below), we observe that our approach, which only uses basic location information as input, outperforms shallow methods that use environmental features. Replacing the input to our method with environmental features only improves performance by +1.1 MAP. This is very encouraging as it indicates that it is possible to learn an environmental feature encoding from raw occurrence data alone that is on par with manually curated feature selection.
>
> **Below we reproduce the results of our comparison against classic SDM baseline, which has been added to the paper.** Both methods are evaluated with either coordinate inputs or environmental feature inputs. All models are trained with 1000 examples per category. Full details can be found in Appendix A1.
>
> | Method | Inputs | S&T |
> |---|---|---|
> | Residual MLP trained with $\mathcal{L}_\mathrm{AN-full}$ | Coordinates | 32.1 |
> | Residual MLP trained with $\mathcal{L}_\mathrm{AN-full}$ | Environmental Covariates | 47.8 |
> | Logistic regression trained with $\mathcal{L}_\mathrm{AN-full}$ | Coordinates | 84.8 |
> | Logistic regression trained with $\mathcal{L}_\mathrm{AN-full}$ | Environmental Covariates | 85.9 |
>
> For reference, we list the methods considered in our paper below.
>
> | Method Name | Reference | Encoder Architecture | Description |
> |---|---|---|---|
> | $\mathcal{L}_\mathrm{AN-full}$ | Ours | Residual MLP |  Training with pseudo-negatives at the data locations and random locations. |
> | $\mathcal{L}_\mathrm{AN-SSDL}$ | Ours | Residual MLP |  Simpler version of $\mathcal{L}_\mathrm{AN-full}$ that only uses pseudo-negatives at random locations. |
> | $\mathcal{L}_\mathrm{AN-SLDS}$ | Ours | Residual MLP |  Simpler version of $\mathcal{L}_\mathrm{AN-full}$ that only uses pseudo-negatives at data locations. |
> | $\mathcal{L}_\mathrm{ME-full}$ | Zhou et al. 2022  | Residual MLP | Merging $\mathcal{L}_\mathrm{AN-full}$ with a state-of-the art SPML loss. |
> | $\mathcal{L}_\mathrm{ME-SSDL}$ | Zhou et al. 2022  | Residual MLP | Merging $\mathcal{L}_\mathrm{AN-SSDL}$ with a state-of-the art SPML loss. |
> | $\mathcal{L}_\mathrm{ME-SLDS}$ | Zhou et al. 2022  | Residual MLP | Merging $\mathcal{L}_\mathrm{AN-SLDS}$ with a state-of-the art SPML loss. |
> | $\mathcal{L}_\mathrm{GP}$ | Mac Aodha et al. 2019 | Residual MLP | $\mathcal{L}_\mathrm{AN-full}$ with additional terms for a user embedding loss. |
> | Best Discretized Grid | Berg et al. 2014 | N/A | Non-parametric discretization of training data, optimized over spatial bin size. |
> | $\mathcal{L}_\mathrm{AN-full}$ (LR, Coord. Inputs) | Pearce et al. 2000 | N/A | Logistic regression using $\mathcal{L}_\mathrm{AN-full}$ with coordinates as input. |
> | $\mathcal{L}_\mathrm{AN-full}$ (LR, Env. Inputs) | Pearce et al. 2000 | N/A | Logistic regression using $\mathcal{L}_\mathrm{AN-full}$ with environmental covariates as input. | |
> | $\mathcal{L}_\mathrm{AN-full}$ (different depths, residual MLP) | Ours | Shallow Residual MLP | Training with pseudo-negatives at the data locations and random locations with shallower variants of our encoder architecture. ||
> | $\mathcal{L}_\mathrm{AN-full}$ (different depths, ordinary MLP) | Ours | Shallow Ordinary MLP | Training with pseudo-negatives at the data locations and random locations with shallower variants of an ordinary MLP. |

---

> ### Author Response · Authors · 2022-11-15
> **General Comments (3/4)**
>
> # C3. Experimental analysis and insight [eJKM, f3dW]
>
> We perform detailed experimental evaluation in the submitted paper. A summary of our findings was presented in our results discussion (Section 4.5 in the submitted version). We have expanded and refined our discussion of results in Section 4.4 of the updated paper. The original submission supported the following key observations:
> 1. Treating species range estimation as a representation learning problem leads to improved performance as a result of training with more data and with more species (Figure 2). These are important insights for the large community interested in range mapping. To the best of our knowledge, we are the first to show the existence of this behavior at scale.
> 2. Methods trained on more data learn more spatially refined representations of the environment (Figure 4).
> 3. Existing state-of-the-art methods for single positive multi-label learning are either not applicable (ROLE in Cole et al. 2020) or not effective (Entropy Maximization in Zhou et al. 2022) on our benchmarks, which novel for the SPML community.
> 4. Unlike Mac Aodha et al. 2019, we separately analyze the role of different types of pseudo-negatives (SSDL vs. SLDS).
> 5. Low-shot performance is surprisingly good. With as few as 10 examples per category (1.3% of the training data), we are able to beat discretized grid methods that use the entire training set (Berg et al. 2014).
>
> We have updated the text to clarify these observations.
>
> We have also added more discussion and analysis in the updated version, including:
> 1. Environmental features are not necessary for good performance. Our coordinate-based approach vastly outperforms a traditional SDM based on environmental features (over +50 MAP). When we use environmental features as the input to our method, the performance gain is small (+1.1 MAP), indicating that our method captures much of the important information without needing environmental features as input. We have also added a discussion of the role of environmental covariates vs. coordinates in Appendix B.2.
> 2. Deep location encoders are crucial for good performance. In Table 1 we can see that shallower models (i.e. logistic regression) do not perform as well as deeper models. In Figure A2 we show that performance increases rapidly with encoder depth.
> 3. **To provide better insight into our models, we have also provided an anonymized tool to explore the model results in detail.** The interactive tool can be used to view detailed model predictions for each species in the *S&T* task. This include precision-recall curves and visualizations of the evaluation data
> On this website it is possible to view predictions for each of the species in the S&T evaluation dataset for the XXX model from Table 1. To minimize hosting costs, the link to this tool is provided in a separate post that is visible only to reviewers. After acceptance, similar detailed per-species results will be made public.
>
> **Note also that we updated our procedures for the *Geo Feature* downstream task.** We found that the default input data normalization in sklearn’s RidgeCV was leading to unstable behavior (https://github.com/scikit-learn/scikit-learn/issues/22540), and that simple min-max scaling of the input features solved the problem. The results for *Geo Feature* now follow similar trends as the other three tasks.

---

> ### Author Response · Authors · 2022-11-15
> **General Comments (4/4)**
>
> # C4. Model architecture and justification [Jszv, apN4, eJKM]
>
> In the submitted paper we illustrated (Figure A.4 in updated paper) and described (Appendix C.1 in updated paper) the network architecture used in our experiments. It is a conventional feed forward architecture consisting of a non-linear projection followed by four residual layers. We choose this network as it was also used in the existing related work of Mac Aodha et al. 2019. **We have updated the text to further clarify the description of the model** to ensure architecture is clear.
>
> **We also ran additional experiments** where we replaced the residual layers with standard fully connected layers and varied the depth of the location encoder. As each residual layer contains two fully connected layers, for this experiment each residual layer was replaced by two fully connected layers to ensure that the models have the same number of total parameters. The results can be found in Figure A.2 and are discussed in Appendix A.2. It turns out that a conventional fully connected network performs comparably to the residual network.

---

> ### Author Response · Authors · 2022-11-15
> **General Comments (1/4)**
>
> # C1. Machine learning novelty and relevance to ICLR [Jszv, apN4, f3dW]
>
> The primary contributions of our work are: (i) showing for the first time that large-scale joint species range mapping from sparse input data is possible, (ii) providing detailed experimental results and drawing connections to related machine learning problem settings (e.g. implicit networks and multi-label learning), and (iii) providing a new benchmark suite of tasks for evaluating large-scale range mapping and environmental representation learning. The importance of the last contribution is worth emphasizing. Our four standardized evaluation tasks will enable machine learning practitioners to develop and directly compare new methods for predicting species ranges from sparse presence-only data. For comparison, one of the most widely cited evaluation datasets from the ecology literature for this task only contains data for 225 species, localized in small geographical regions [1]. In addition, all of their species are anonymized and the raw location information for half of their species has been obfuscated for conservation purposes, making it impossible to evaluate on their data directly. For context, a recent study using the dataset from [1] found little spread between traditional SDM methods (most falling between 0.69 and 0.74 AUROC), and showed that even penalized linear regression baselines were competitive [6].
>
> We believe that our work is well within scope for ICLR. Our most relevant subject area is “Machine Learning for Sciences (eg biology, physics, health sciences, social sciences, climate/sustainability)” which indicates that the conference is open to works such as ours. In addition, the ICLR call for papers (https://iclr.cc/Conferences/2023/CallForPapers) specifically includes “climate, sustainability” and “applications in audio, speech, robotics, neuroscience, biology, or any other field” as relevant topics.
>
> Furthermore, there are numerous examples of prior accepted ICLR papers (including orals) whose primary contribution is not a methodological advancement, but a new benchmark suite or datasets (e.g. [2, 3, 4, 5] from ICLR 2022 alone). Benchmark datasets such as the ones we present are precursors to new methods and advancements. The problem domain we explore (i.e. predicting species range maps) has important consequences for mitigating biodiversity losses due to climate change. As a result, there is a large potential net benefit of making this problem more accessible to machine learning researchers.
>
> [1] Elith et al. (Ecography 2006). Novel methods improve prediction of species’ distributions from occurrence data.
> [2] Sagawa et al. (ICLR 2022). Extending the WILDS Benchmark for Unsupervised Adaptation.
> [3] Zela et al. (ICLR 2022). Surrogate NAS Benchmarks: Going Beyond the Limited Search Spaces of Tabular NAS Benchmarks.
> [4] Sixt et al. (ICLR 2022). Do Users Benefit From Interpretable Vision? A User Study, Baseline, And Dataset.
> [5] Liang et al. (ICLR 2022). MetaShift: A Dataset of Datasets for Evaluating Contextual Distribution Shifts and Training Conflicts.
> [6] Valavi et al. (Ecological Monographs 2022). Predictive performance of presence-only species distribution models: a benchmark study with reproducible code.

---

### Decision · Program_Chairs · 2023-01-20

**Decision:**

Reject

**Justification For Why Not Higher Score:**

The paper quality is clearly below the bar of ICLR and the representation learning part is not enough in the paper.

**Justification For Why Not Lower Score:**

N/A

**Metareview: Summary, Strengths And Weaknesses:**

This is an application paper working on species distribution modeling. The paper proposed to learn from not very reliable and not very costly data called crowdsourced data (which seem not necessarily collected by crowdsourcing). While the studied problem is interesting and important in natural science, the paper quality is clearly below the bar of ICLR and the representation learning part is not enough in the paper. Thus, we cannot accept it for publication. If the authors would like to revise the paper and try another machine learning conference, I suggest the authors to carefully consider what would be novel from the machine learning point of view in the revised manuscript.